# Can Graph Learning Improve Planning in LLM-based Agents?

**Xixi Wu**[1,3*]   **Yifei Shen**[2*✉]   **Caihua Shan**[2]   **Kaitao Song**[2]   **Siwei Wang**[2]   **Bohang Zhang**[4]
**Jiarui Feng**[5]   **Hong Cheng**[3]   **Wei Chen**[2]   **Yun Xiong**[1✉]   **Dongsheng Li**[2]

[1]Fudan University[†]   [2]Microsoft Research Asia   [3]The Chinese University of Hong Kong
[4]Peking University   [5]Washington University, Saint Louis

## Abstract

Task planning in language agents is emerging as an important research topic alongside the development of large language models (LLMs). It aims to break down complex user requests in natural language into solvable sub-tasks, thereby fulfilling the original requests. In this context, the sub-tasks can be naturally viewed as a graph, where the nodes represent the sub-tasks, and the edges denote the dependencies among them. Consequently, task planning is a decision-making problem that involves selecting a connected path or subgraph within the corresponding graph and invoking it. In this paper, we explore graph learning-based methods for task planning, a direction that is orthogonal to the prevalent focus on prompt design. Our interest in graph learning stems from a theoretical discovery: the biases of attention and auto-regressive loss impede LLMs' ability to effectively navigate decision-making on graphs, which is adeptly addressed by graph neural networks (GNNs). This theoretical insight led us to integrate GNNs with LLMs to enhance overall performance. Extensive experiments demonstrate that GNN-based methods surpass existing solutions even without training, and minimal training can further enhance their performance. The performance gain increases with a larger task graph size. [3]

## 1   Introduction

LLM-based agents have recently emerged as a rapidly growing field of research and are considered a significant step towards artificial general intelligence (AGI) [61, 7]. These agents have achieved remarkable successes across a variety of domains, as evidenced by their ability to address complex AI challenges (e.g., HuggingGPT [46]), excel in gaming environments (e.g., Voyager [58]), and drive innovation in chemical research (e.g., [5]). Within this burgeoning field, task planning in language agents emerges as a critical area of study. It involves LLMs autonomously interpreting user instructions, breaking user's instructions in natural language into concrete and solvable sub-tasks, and then fulfilling the user's request by executing each sub-task [43, 46, 45]. For instance, in the case of HuggingGPT [46], task planning involves invoking expert AI models from the HuggingFace website to solve complex AI tasks beyond the capabilities of GPT alone.

Given its practical significance, numerous algorithms have been proposed, with a major focus on prompt design [45, 46, 33, 30, 60, 49, 71, 4, 63]. This paper proposes to explore an orthogonal

---

[*] denotes equal contributions. Work was done during Xixi Wu's (xxwu@se.cuhk.edu.hk) internship at Microsoft Research Asia. Corresponding authors (yifeishen@microsoft.com, yunx@fudan.edu.cn)

[†] Shanghai Key Laboratory of Data Science, School of Computer Science, Fudan University

[3] The code and datasets are available at https://github.com/WxxShirley/GNN4TaskPlan

direction, i.e., graph-learning-based approaches. In task planning, solvable sub-tasks can be naturally represented as a *task graph*, wherein each node corresponds to a distinct sub-task, and each edge signifies the dependencies between these sub-tasks. The crux of task planning, therefore, involves selecting a connected path or subgraph to satisfy the user's request, which is a decision-making problem on graphs. Adopting this framework, we analyze the task planning capabilities of LLMs, specifically within the context of HuggingGPT [46]. Our empirical investigation uncovers that a considerable portion of planning failures can be ascribed to the LLMs' inefficacy in accurately discerning the structure of the task graph. This finding presents intriguing questions from both theoretical and empirical perspectives. Theoretically, it initiates a discussion on the inherent limitations of LLMs in processing task graphs. Empirically, it highlights the urgent need for developing effective and efficient strategies to mitigate this deficiency and improve task planning performance.

For the theoretical question, we first investigate the expressiveness of Transformer architectures when applied to graph tasks with sequential graph input, such as edge list representations, which is the graph input format for task planning. Our initial hypothesis is that the format of sequential graph input might not align with the inductive bias inherent to graph structures, potentially reducing expressiveness. Contrary to this hypothesis, it is proved that by taking edge lists as the input, a constant-width Transformer can solve graph decision-making problems by simulating dynamic programming algorithms on edge lists. Nevertheless, we find that LLMs' solutions lack invariance under graph isomorphism, an important property for graph decision-making problems. In addition, the expressiveness is weakened if the attention is sparse, which is typically observed in LLMs [66]. Beyond expressiveness, we also examine the influence of auto-regressive loss, demonstrating that it introduces spurious correlations that can be harmful to graph decision-making tasks. These insights expose the inherent limitations of LLMs in task planning and, more broadly, in graph-related problems (e.g., the challenges in [14, 59, 34]).

To tackle the limitations, we take the use of GNNs, which have been shown to adeptly handle graph decision-making problems, both in theory and in practice [24, 68]. Initially, we deploy LLMs to interpret an ambiguous user request, breaking it down into more detailed steps. Subsequently, we utilize a GNN to retrieve the relevant sub-tasks based on these detailed steps and the corresponding sub-task descriptions. Notably, this approach can be implemented without training if we adopt parameter-free GNN models such as SGC [65]. In the case of training-based methods, we apply the Bayesian Personalized Ranking (BPR) loss [41] to facilitate learning from the implicit sub-task rankings. Extensive experiments demonstrate that the proposed methods achieve better performance than baselines. Specifically, our main contributions are summarized as follows:

1. **Task Planning Formulation:** This study presents a formulation of task planning as a graph decision-making problem. In the realm of task planning, our work initiates the exploration of graph learning methodologies to enhance performance. Concurrently, it introduces task planning as a new application in the graph learning domain.

2. **Theoretical Insights:** We prove that Transformers have expressiveness to solve graph decision-making problems *based on edge list input*, but inductive biases of attention and the auto-regressive loss function may serve as obstacles to their full potential.

3. **Novel Algorithms:** Based on the theoretical analysis, we introduce an additional GNN for sub-task retrieval, available in both training-free and training-based variants. The experiments on diverse LLMs and planning datasets demonstrate that the proposed method outperforms existing solutions with much less computation time. Furthermore, the performance is further enhanced by improved prompts or a fine-tuned model.

## 2 Preliminaries

In this section, we introduce task planning in language agents and the current LLM-based solutions.

### 2.1 Task Planning in Language Agents

We start with the definition of task planning with a concrete example of HuggingGPT [46]. In task planning, there is a pool of pre-defined tasks. Task planning inputs include this task pool and a user request. The user request is expressed in natural language, which is ambiguous and could encompass

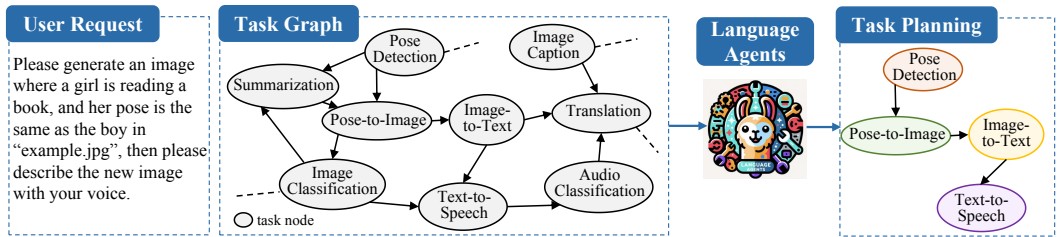

Figure 1: **Illustration of Task Planning in Language Agents (e.g., HuggingGPT [46])**

multiple complex tasks. The output is a sequence of tasks and the order of their invocation to address the user's request.

Figure 1 features the task planning in HuggingGPT, with the pre-defined tasks corresponding to APIs from the HuggingFace website, such as `Translation` and `Pose-to-Image`, accompanied by detailed descriptions. For instance, the detailed description for `Translation` is "Translation is the task of converting text from one language to another". The user request is "Please generate an image where a girl is reading a book, and her pose matches the boy in 'example.jpg', then describe the new image with your voice." The ground-truth output is a sequence of four APIs (nodes): {`Pose Detection`, `Pose-to-Image`, `Image-to-Text`, `Text-to-Speech`}, outlining the order of their invocation (a path). By invoking these APIs on HuggingFace, the user request can be fulfilled.

## 2.2  Current LLM-based Solution to Task Planning

The current solution of task planning is purely based on LLMs and involves two stages [45, 46]. The first stage involves request decomposition, where a user's ambiguous request is broken down into concrete *steps* via LLMs. For instance, the request illustrated in Figure 1 is decomposed into the following steps: (1) analyze the pose of the boy; (2) take that pose and generate a new image; (3) generate the caption for newly generated image; (4) convert the generated text into audio. The second stage is task retrieval. For each decomposed step, LLMs are employed to retrieve an appropriate task from the task pool and execute them in sequence. For example, "Analyze the pose of the boy" corresponds to `Pose detection`. The output tasks should be (1) `Pose detection`; (2) `Pose-to-Image`; (3) `Image-to-Text`; (4) `Text-to-Speech`. Figure 6 illustrates this procedure.

## 3  Graph Formulation and Insights

### 3.1  Graph Formulation of Task Planning

In this subsection, we formulate the task planning as a decision-making problem on the task graph. The task graph is a special kind of text-attributed graphs and we define it as $G = (V, E, T)$. Each node $v \in V$ represents a pre-defined task in the task pool, associated with a text $t_v \in T$ describing its function (e.g., "Translation. Translation is the task of converting text from one language to another."). Each edge $(u, v) \in E$ indicates a dependency between tasks (e.g., the output format of task $u$ matches the input format of task $v$). Task planning is to select a path or connected sub-graph on the task graph.

Viewed from this angle, task planning bears resemblance to traditional decision-making problems on graphs, such as planning for the shortest path. Compared with traditional planning, task planning in language agents involves diverse and open-ended goals due to the varied users' personal requests. For example, on platforms like HuggingFace, users' intentions span across video, text, and image domains. On the contrary, classic planning has a fixed goal for a given domain, which is often explicitly expressed by mathematical formulas [55, 36].

### 3.2  Failures of LLMs in Planning: Empirical Findings

With the task graph at hand, we diagnose LLMs in task planning in Figure 2. We adopt the experimental settings as outlined in the work of HuggingGPT [46], where the prompts are specifically optimized for task planning on HuggingFace. The evaluation metric calculates the F1 score to assess the accuracy of the tasks identified by LLMs against the ground-truth tasks. Additionally, we report

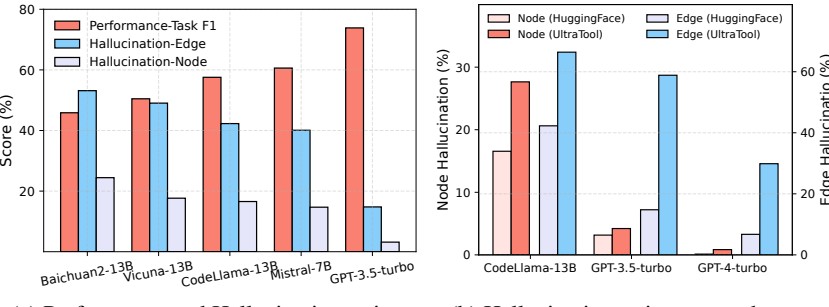

(a) Performance and Hallucination ratios      (b) Hallucination ratios across datasets

Figure 2: **Illustration of (a) LLMs' planning performance and hallucination in HuggingGPT, and (b) hallucination in relation to task graph size.**

two task-graph-related metrics: the node hallucination ratio and the edge hallucination ratio. These metrics measure the frequency of non-existent nodes (i.e., tasks) and edges (i.e., dependencies) outputted by LLMs, respectively, indicative of the models' misinterpretation of the graph input.

Our empirical findings reveal that (1) LLMs exhibit a certain **hallucination** ratio, and (2) there is a strong correlation between the hallucination ratio and planning performance. This suggests that LLMs struggle to accurately interpret the task graph while the task graph is the key to the performance.

We further explore whether the incidence of hallucinations correlates with the number of sub-tasks. The HuggingGPT dataset contains 23 sub-tasks, and our analysis is expanded to incorporate the UltraTool dataset [20], which consists of 260 sub-tasks. Figure 2b illustrates that the hallucination ratio increases with a larger task graph size.

## 3.3 Failures of LLMs in Planning: Theoretical Insights

In this subsection, we provide theoretical insights into the limitations of LLMs in processing task graphs. In contrast to previous graph learning approaches for graph decision-making problems, LLMs process the graph input by flattening it into a sequence and are trained using an auto-regressive loss. We will then examine the impact of these two factors.

**How does sequential graph input impact the expressiveness?** We consider general graph decision-making problems that can be resolved using dynamic programming (DP) as described in (2). The input comprises the edge list and initial states:

$$\underbrace{u_1 \ v_1 \ c[u_1][v_1] \ u_1 \ v_2 \ c[u_1][v_2] \ \dots}_{\text{edge list}} \ \underbrace{u_1 \ \text{Answer}[0][u_1] \ \dots \ u_n \ \text{Answer}[0][u_n]}_{\text{initial states}} \qquad (1)$$

The intended output format is $u_1 \ \text{Answer}[k][u_1] \ \dots \ u_n \ \text{Answer}[k][u_n]$. In existing studies, task graphs are often presented in (1), where the edge list is described by natural language and the initial states are the task descriptions of each task node, detailed in Appendix A.9 of [45].

As discussed in the previous subsection, task planning is a decision-making problem on the task graph. The decision-making problems on graphs are often solved by DP [3] and its general formulation is given by

$$\text{Answer}[k][i] = f\left(\square_{j \in \mathcal{T}(i)} g(\text{Answer}[k-1][j], c[i][j])\right), \qquad (2)$$

where $\text{Answer}[k][i]$ is the solution to state $i$ in the $k$-th iteration, $c[i][j]$ is a cost associated with state $i$ and $j$, $\mathcal{T}(i)$ is the set of state can be transited to $i$, $\square_{j \in \mathcal{T}(i)}$ is an aggregation function such as MAX or SUM, and $f, g$ are task-specific update functions. We give the formulation of typical DP algorithms in Appendix D.1 including some NPC problems. For the decision-making problems on the text-attributed graphs (e.g., task planning), one may conceptualize them as DP in the feature space, as discussed in [68].

To our surprise, although the edge list input does not directly reflect the geometric structures of graphs, it enables Transformers to simulate DP efficiently, in terms of expressiveness, as demonstrated by the following theorem.

**Theorem 1.** *(LLMs have enough expressiveness) Assume the input format is given in (1) and $f, g, \square$ in DP update (2) satisfy the assumptions 1 and 2 in Appendix. There exists a log-precision constant-depth and constant-width Transformer that simulates one step of DP update in (2). As a consequence, there exists a log-precision $O(k)$-depth and constant-width Transformer that simulates $k$ steps of DP update in (2).*

The proof is presented in Appendix D.2. However, certain aspects of the proof's constructions are challenging to be realized in Transformers that have been pretrained on natural language. First, the embedding process must be carefully filtered to ensure invariance under graph isomorphism. This invariance property does not align with the inductive biases inherent in natural language, making it difficult to achieve. Consequently, if an LLM can accurately produce the correct answer for a specific ordering of nodes, it might not maintain this accuracy after the nodes have been reordered (experiments given in Appendix D.3). Second, each token needs to synchronize its hidden states with all other tokens sharing the same token ID, which is of order $O(|V|)$. In practice, the attention trained from natural language is typically sparse [66], leading to intractability issues. The formal lower bound is provided in the following proposition and the proof is given in Appendix D.4.

**Proposition 1.** *(Inductive bias of language hinders expressiveness) Assume the input format is described (1) and that the attention mechanism is limited to attending to a constant number of tokens. There exists at least one instance of one-step DP update such that no log-precision constant-width constant-depth transformer can simulate.*

**How does auto-regressive Loss impact the generalization?** Our investigation next focuses on the auto-regressive loss and considers the following scenario: given a fixed task graph, user data is collected to perform instruction tuning with next-token-prediction loss. For a tractable theoretical study, we conceptualize this issue as a path-planning problem, since the output of task planning is essentially a path. We consider the training dataset comprises input sequences of the form $s\ t\ s\ v_1\ v_2\ \cdots\ t$, where $s$ represents the source node, $t$ the target node, and the sequence $s\ v_1\ v_2\ \cdots\ t$ is a path that adheres to specified constraints. During testing, given the initial and target nodes $s$ and $t$, the model is expected to generate a path with the same constraint. Our findings indicate that auto-regressive loss can lead to the emergence of a frequency-based spurious correlation, as substantiated by the following theorem and the proof is given in Appendix D.5.

**Theorem 2.** *(Spurious correlations of auto-regressive loss) Assume (1) the loss employed is a next-token-prediction loss utilizing cross-entropy, applied to the sub-sequence $v_1\ v_2\ \cdots\ t$ during training; (2) the output logits are determined by target node $t$ and the current node $v_{i-1}$. Let $N_{t,v_{i-1},u}$ be the number of times in the training dataset such that $t$ is the target node, $v_{i-1}$ is the current node and $v_i = u$ is the next node. The optimal logits for predicting the next node $u$ from current node $v_{i-1}$ towards target node $t$ is given by $\hat{v}_i[u] = \frac{N_{t,v_{i-1},u}}{\sum_u N_{t,v_{i-1},u}}$ if $\sum_u N_{t,v_{i-1},u} > 0$. If $\sum_u N_{t,v_{i-1},u} = 0$, $\hat{v}_i[u]$ can be any non-negative number subject to $\sum_u \hat{v}_i[u] = 1$.*

In our setup, $s\ t$ is the instruction and the third token is a duplicate of the first token. It is reasonable to exclude these tokens in the loss calculation, which is the first assumption. The second assumption assumes that the output only depends on the current node and target node, which is a minimal requirement for path-related problems. For DP problems, the frequency-based prediction contradicts to the value-based ground-truth. We then give an example that auto-regressive loss even cannot find a valid path.

**Example 1.** Consider a training dataset consisting of a sufficient number of valid paths. Suppose the dataset contains two paths $a\ b\ c$ and $b\ c\ d$ and there are no other paths such that $t = d$ and the current node $v_i = a$ for all $i$. Then we have $N_{d,a,u} \equiv 0$ for all $u$ and the logits for the next node can be arbitrary. This results in the model's inability to predict the next node of $a$ when given $a$ as the source node and $d$ as the target node.

To a human, finding a path from $a$ to $d$ simply involves concatenating the paths $a\ b\ c$ and $b\ c\ d$. However, auto-regressive loss fails under such circumstances. In task planning datasets, we indeed observe that the performance of fine-tuned LLMs is inferior to that of GNNs trained on the same dataset, as shown in Figure 3b.

# 4 Integrating GNNs and LLMs for Planning

## 4.1 Motivations

In the last section, we find that a considerable portion of planning failures can be ascribed to the LLMs' inefficacy in accurately discerning the structure of the task graph, due to the hallucination, the inductive bias of the attention, and next-token prediction loss. In contrast to LLMs, GNNs can strictly operate on the task graph, thereby avoiding hallucinations. Additionally, they leverage the graph structure as input, rather than flattening the graph into a sequence, thus overcoming the theoretical limitations discussed previously. Furthermore, GNNs have demonstrated proficiency in handling graph decision-making problems, both theoretically and empirically [68, 11, 24]. As a result, the simplest fix is to integrate GNNs into the task-planning algorithm.

In the following subsections, we propose both training-free and training-based approaches to enhance performance. Training-free methods are necessary when the available tasks are continuously changing, or new tasks are emerging constantly. This scenario is common when the task planning module is deployed in a new system. Once the task planning module has been deployed for a period, it becomes possible to collect users' requests and label a small proportion of the data, enabling lightweight training-based methods.

## 4.2 A Training-free GNN-based Approach

As we discussed in Section 2.2, the current solution to task planning involves two stages. The first stage requires the ability to understand users' requests in natural language and break them down into concrete instructions, which is the unique ability of LLMs. The second stage is to select a path on the task graph, where each node corresponds to a decomposed step. Thus, we can integrate GNNs in this stage. The illustration of our framework is shown in **Figure 7 in Appendix**.

For each decomposed step outputted by the first stage, we use a GNN to select a corresponding node within the task graph. Suppose we are selecting the node for the $i$-th decomposed step. First, we utilize a small pre-trained language model, e5-335M [62], to embed the $i$-th decomposed step. The resulting embedding is denoted as $\boldsymbol{x}_i^{\text{step}}$. Second, for the task graph, we first use the same pre-trained language model e5-335M to convert each node's description into embeddings, denoted as the node feature $\boldsymbol{h}_v^0$, where the superscript indicates the layer and the subscript represents the node. Then we adopt a $K$-layer SGC [65] to compute the final node embeddings, resulting in $\boldsymbol{h}_v = \boldsymbol{h}_v^{(K)}$. Given a sequence of previously selected task nodes $\{v_1, \cdots, v_{i-1}\}$, the next node $v_i$ is chosen according to $v_i = \operatorname{argmax}_{v \in \mathcal{N}(v_{i-1})} \langle \boldsymbol{h}_v, \boldsymbol{x}_i^{\text{step}} \rangle$, where $\boldsymbol{h}_v$ is the final node embedding, and $\mathcal{N}(v_{i-1})$ denotes the neighbors of node $v_{i-1}$ in the task graph. Particularly, $v_1$ can be selected from the whole graph.

As e5-335M is pre-trained and SGC is parameter-free, the proposed method requires no additional training and can be effectively applied in a zero-shot manner.

## 4.3 A Training-required GNN-based Approach

The inference process in training-required methods mirrors that of the training-free approach, with the difference being the substitution of parameter-free GNNs with parametric counterparts, such as GAT [57] or GraphSAGE [15]. Here we specify the training process of GNNs.

**Data Preparation:** We assume that each entry in the task planning dataset comprises a user request, a sequence of decomposed steps, and the corresponding ground-truth tasks, denoted as (request, $\{s_1, \ldots, s_n\}$, $\{v_1, \ldots, v_n\}$). If the dataset does not adhere to this format, we reformat it accordingly using GPT-4, with details provided in Appendix C.2. It is important to note that there is a one-to-one correspondence between the steps and tasks in the dataset. Therefore, the training dataset can be represented as $\{(s_i, v_i)\}_{i=1}^n$, where $s_i$ is a step described in natural language, and $v_i$ is its corresponding invoked task.

**Training Loss:** The problem in the dataset can be viewed as a binary ranking problem, where the labeled node is 1 and the other nodes are 0. Therefore, we adopt the Bayesian Personalized Ranking (BPR) loss [41] designed for recommendation with binary rankings. The loss function is given by $\ell = \sum_{(\boldsymbol{x}^{\text{step}}, v, v')} - \log \ \sigma(\langle \boldsymbol{h}_v, \boldsymbol{x}^{\text{step}} \rangle - \langle \boldsymbol{h}_{v'}, \boldsymbol{x}^{\text{step}} \rangle)$, where $\boldsymbol{x}^{\text{step}}$ represents the embedding of the step's textual description generated by e5-335M, $v$ is the ground-truth task, and $v'$ is a negative task.

We select negative tasks that are textually similar to the positive task, and for computational efficiency, we limit our selection to 2 negative tasks per positive task. The trainable parameters may merely include GNNs or both GNNs and e5-335M with illustrative configurations shown in **Figure 8 in Appendix**.

## 5 Experiments and Analysis

### 5.1 Experimental Setup

**Datasets:** We utilize four datasets across two task planning benchmarks: HuggingFace tasks, Multimedia tasks, and Daily Life API tasks from **TaskBench** [45], as well as TMDB API tasks from **RestBench** [50]. The HuggingFace dataset includes AI models on the HuggingFace. The Multimedia dataset provides a wide range of user-centric tasks, such as file downloading and video editing. The Daily Life APIs cater to everyday services like web search and shopping functionalities. TMDB focuses on movie-related search and retrieval tasks. Statistics for each dataset are presented in Table 7 with illustrative examples shown in Figure 4 in Appendix. Other benchmarks, such as ToolBench [39] and ToolAlpaca [54], are less suitable for our experiments due to (1) the absence of a well-defined task graph detailing tasks and their dependencies, and (2) a scarcity of samples involving multi-task planning, with a focus on single-task retrieval.

**Evaluation:** For the datasets from TaskBench, we split 3000 samples for training and 500 samples for testing, each containing an invocation path with at least two tasks. For the TMDB dataset, we first filter to include the samples with two or more invoked tasks, and then randomly select a sample served as the in-context learning example. The remaining 94 samples are designated for testing. We adopt the evaluation metric in TaskBench [45] and HuggingGPT [46], i.e., Node F1-Score (*n-F1*) and Link F1-Score (*l-F1*), which measure the accuracy of invoked tasks and invoked dependencies, respectively. Besides, the Accuracy (*Acc*) can measure the success rate from task level. We also measure the token consumption (*# tok*) as the efficiency metric.

**Choices of LLMs:** We consider close-sourced LLMs, i.e., GPT-3.5-turbo and GPT-4-turbo, as well as open-sourced LLMs with different parameter scales, including CodeLlama-13B(or 7B)-Instruct-hf [42], Mistral-7B-Instruct-v0.2 [23], Vicuna-13B-v1.5 [72], and Baichuan2-13B-Chat [69].

**Choices of GNNs:** To comprehensively investigate the effectiveness of different graph learning methods for task planning, we consider a wide range of graph neural networks, including SGC [65], GCN [26], GAT [57], GraphSAGE [15], GIN [67], and Graph Transformers [47].

### 5.2 Performance of the Training-free Approach

We compare the performance across three training-free methods: **(1) LLM's Direct Inference** is introduced in Section 2.2. **(2) GraphSearch** [33, 52, 32] leverages the classic graph search method to generate the candidate nodes and uses LLMs to give a score for node selection. Given a step, **GreedySearch** consistently selects the node with the highest score and adjacent to the previous task node; **AdaptiveSearch** selects the nodes with scores above a fixed threshold, adjusting the breadth of the search space in an adaptive mode; **BeamSearch** retains the $k$ nodes with highest scores. **(3) SGC** [65] employs a training-free SGC for task retrieval based on decomposed task steps. The details of baselines are given in Section E.1 and illustrated in Figure 6. Table 1 shows both the overall performance and token consumption costs, with results of Accuracy (*Acc*) moved to Table 9 in Appendix.

Compared with direct inference, integrating an SGC consistently improves performance, underscoring the effectiveness of the proposed method. GraphSearch-type methods rely on beam search to identify paths and employ LLMs for evaluation, where longer processing times generally lead to better outcomes. Notably, our proposed method achieves **comparable or superior performance** (Table 1 and Table 9) to BeamSearch while requiring 5-10 **times fewer tokens** (Table 1) and inference time (Table 10). The case studies are provided in Appendix H. However, we observed only marginal improvements with GPT-4-turbo. A unique feature of GPT-4-turbo is its ability to manage ChatGPT-plugins, and it may have been specially trained on task planning datasets. In addition, the pre-trained language model used for feature extraction in SGC is e5-335M, which may not be sufficiently powerful to effectively analyze GPT-4's output. A detailed diagnostic analysis of cases involving GPT-4 is provided in Figure 11 in Appendix.

Table 1: **Comparison of Training-free Methods: Overall Performance (Node-F1 and Link-F1 in %) and Token Consumption in** $\times 10^3$. Performance of other LLMs are given in Table 8.

| LLM | Method | HuggingFace | | | TaskBench Multimedia | | | Daily Life | | | RestBench TMDB | | |
|---|---|---|---|---|---|---|---|---|---|---|---|---|---|
| | | n-F1 ↑ | l-F1 ↑ | # Tok ↓ | n-F1 ↑ | l-F1 ↑ | # Tok ↓ | n-F1 ↑ | l-F1 ↑ | # Tok ↓ | n-F1 ↑ | l-F1 ↑ | #Tok ↓ |
| **Vicuna 13B** | Direct | 50.46 | 21.27 | 2.50 | 53.57 | 23.19 | 2.64 | 73.70 | 45.80 | 3.82 | 44.66 | 14.01 | 2.02 |
| | GreedySearch | 52.94 | 25.73 | 6.23 | 46.99 | 23.11 | 5.55 | 42.98 | 13.33 | 7.18 | 45.22 | 13.69 | 3.42 |
| | AdaptiveSearch | 54.36 | 25.67 | 9.81 | 51.24 | 24.32 | 11.25 | 62.71 | 31.15 | 13.92 | 41.32 | 7.02 | 6.51 |
| | BeamSearch | 56.64 | 26.93 | 24.11 | 54.09 | 26.19 | 25.42 | 54.55 | 23.60 | 24.86 | 46.91 | 15.41 | 7.79 |
| | SGC | **59.62** | **31.98** | **2.31** | **61.78** | **37.60** | **2.43** | **83.33** | **53.31** | **3.82** | **48.79** | **15.99** | **1.89** |
| **Mistral 7B** | Direct | 60.60 | 30.23 | 2.49 | 69.83 | 39.85 | 2.64 | 84.26 | 53.63 | 3.77 | 62.23 | 22.02 | 1.96 |
| | GreedySearch | 65.91 | 38.13 | 6.52 | 58.92 | 34.72 | 6.26 | 75.18 | 49.47 | 8.27 | 60.64 | 23.18 | 4.38 |
| | AdaptiveSearch | 67.30 | 38.90 | 7.68 | 71.59 | 44.84 | 10.66 | 86.39 | 63.65 | 10.92 | 54.04 | 21.35 | 9.99 |
| | BeamSearch | 67.13 | 36.73 | 25.66 | 73.55 | 47.12 | 31.10 | 85.87 | 61.53 | 39.16 | 63.41 | **26.79** | 11.26 |
| | SGC | **67.43** | **42.08** | **2.32** | **74.07** | **49.90** | **2.43** | **87.13** | **66.49** | **3.54** | **64.72** | 25.67 | **1.89** |
| **CodeLlama 13B** | Direct | 57.55 | 28.88 | 2.45 | 68.57 | 41.79 | 2.59 | 91.20 | 76.07 | 3.88 | 68.91 | 43.74 | 2.02 |
| | GreedySearch | 61.67 | 34.02 | 5.95 | 67.98 | 42.04 | 4.95 | 91.50 | 76.56 | 5.54 | 66.67 | 42.16 | 3.81 |
| | AdaptiveSearch | 60.85 | 31.66 | 11.10 | 68.14 | 41.71 | 6.77 | 91.34 | 76.09 | 7.18 | 63.74 | 37.17 | 8.16 |
| | BeamSearch | 62.65 | 34.31 | 20.14 | 69.53 | 43.35 | 19.51 | 91.74 | 76.60 | 19.19 | 68.08 | 42.92 | 8.88 |
| | SGC | **65.51** | **39.44** | **2.31** | **73.32** | **53.28** | **2.43** | **92.96** | **79.57** | **3.64** | **71.40** | **47.55** | **1.90** |
| **GPT-3.5-turbo** | Direct | 73.85 | 45.73 | 2.14 | 82.85 | 62.07 | 2.26 | 96.09 | 83.65 | 3.36 | 81.70 | 57.52 | 1.67 |
| | GreedySearch | 67.75 | 43.88 | 5.29 | 81.11 | 63.02 | 4.92 | 93.77 | 81.26 | 7.36 | 76.19 | 50.11 | 3.06 |
| | AdaptiveSearch | 72.18 | 47.55 | 7.47 | 81.86 | 62.71 | 5.71 | 93.79 | 81.41 | 8.53 | 77.57 | 53.65 | 5.89 |
| | BeamSearch | 75.51 | 49.62 | 14.22 | 83.57 | **64.50** | 12.91 | 95.66 | 82.72 | 22.05 | 81.24 | 57.98 | 6.42 |
| | SGC | **76.37** | **50.04** | **2.02** | **83.65** | 63.65 | **2.09** | **96.38** | **86.19** | **3.16** | **82.63** | **59.15** | **1.61** |
| **GPT-4-turbo** | Direct | 77.60 | 52.18 | 2.19 | 88.29 | 69.38 | 2.28 | **97.36** | 84.58 | 3.37 | **82.56** | **56.67** | 1.75 |
| | GreedySearch | 74.75 | 50.44 | 5.78 | 86.81 | 69.80 | 5.52 | 97.36 | 85.78 | 7.37 | 75.34 | 49.95 | 3.73 |
| | AdaptiveSearch | 76.17 | 51.30 | 8.94 | 88.02 | 69.99 | 7.14 | 97.30 | **85.80** | 9.04 | 81.78 | 55.15 | 6.35 |
| | BeamSearch | 77.56 | **52.54** | 8.98 | 88.16 | **70.39** | 6.90 | 97.35 | 85.78 | 8.99 | 80.11 | 51.00 | 5.18 |
| | SGC | **77.79** | 52.20 | **2.03** | **88.54** | 69.83 | **2.10** | 97.35 | 85.76 | **3.16** | 82.27 | 56.37 | **1.62** |

## 5.3 Performance of the Training-based Approaches

**Settings:** We further explore the efficacy of training-based GNNs in three TaskBench datasets. The TMDB dataset is excluded due to its limited sample size. Throughout our experiments, we trained a spectrum of GNN variants, both with and without co-training the small LM (i.e., e5-335M), whose role is to generate node embeddings derived from task names and descriptions. Owing to space constraints, we only show the performance of GraphSAGE in the main text, relegating a detailed comparison of all the situations to Table 11 and Table 12 in Appendix.

**Compared Methods:** Due to the lack of training-based baseline methods specifically for task planning, we adapt two existing approaches that combine LLMs and GNNs for graph-related tasks, including: (1) **TAPE** [16] employs a LLM → LM → GNN architecture for node classification task. In this framework, LLMs generate high-quality explanatory text for each predicted node, which is then fine-tuned by an LM to produce node embeddings. Finally, a GNN performs the downstream classification. We adapt TAPE for task planning by reformulating the problem as classifying user requests into corresponding node labels within the task graph. (2) **GraphToken** [38] uses GNNs to tokenize graph nodes, which are then fed into LLMs to generate textual outputs. In our adaptation for task planning, we treat the user request as the input question and the expected plan as the generated answer. Additional implementation details are provided in the Appendix F.2.

**Observations:** From Table 2, we observe a significant improvement in performance when employing a training-based GraphSAGE approach over the training-free method. However, the co-training of GNNs with e5-335M does not yield a marked improvement, suggesting that message passing is the crucial element for enhancing performance. Further analysis across a broad spectrum of GNNs (as shown in Table 11 and Table 12) reveals that powerful GNNs, such as GINs, perform similarly to networks perceived as less complex, like GCNs, and even underperform compared to GraphSAGE. This pattern indicates that the task's challenge may not lie in the expressiveness of the models but rather in their ability to generalize. Regarding baselines, TAPE is unsuitable for task planning as its classification approach simplifies task planning, overlooking task dependencies. While GraphToken demonstrates superior performance over LLMs' direct inference, we have noted instances of minor hallucination. This observation suggests that GraphToken's understanding of the task graph is not yet perfect. In addition, GraphToken is limited to open-sourced LLMs. Besides, our proposed approaches also greatly boost the parameter prediction performance, as given in Appendix G (e.g. 9% improvement to GPT-3.5-turbo and 3% improvement to GPT-4-turbo).

**Efficiency:** The details of the training time are given in Table 13. The training cost is remarkably low because we use e5-335M [62] as the text embedding model for GNNs. If the trainable parameters are limited to the GNNs alone, training typically concludes within just 3 minutes. Furthermore, the

Table 2: **Comparison with Training-based Approaches: Node-F1, Link-F1, and Accuracy are reported in** %. TAPE [16] is designed for node classification task and cannot predict links, so we report Link-F1 as "NA". GraphToken [38] requires finetuning LLMs, which is not compatible with close-sourced LLMs. The performance of other GNNs and LLMs are given in Table 11 and Table 12 in the Appendix.

| LLM | Method | HuggingFace | | | Multimedia | | | Daily Life | | |
|---|---|---|---|---|---|---|---|---|---|---|
| | | n-F1 ↑ | l-F1 ↑ | Acc ↑ | n-F1 ↑ | l-F1 ↑ | Acc ↑ | n-F1 ↑ | l-F1 ↑ | Acc ↑ |
| **Vicuna-13B** | Direct | 50.46 | 21.27 | 8.72 | 53.57 | 23.19 | 11.20 | 73.70 | 45.80 | 24.43 |
| | TAPE | 59.47 | NA | 5.07 | 54.97 | NA | 2.07 | 73.26 | NA | 12.50 |
| | GraphToken | **63.37** | 31.54 | 15.61 | 65.40 | 36.38 | 19.87 | 81.65 | 48.29 | 43.37 |
| | GraphSAGE | 61.86 | 35.68 | **20.08** | 63.71 | 39.88 | 21.37 | **86.07** | **67.63** | **48.64** |
| | GraphSAGE$_{co\text{-}train}$ | 62.82 | **37.04** | 19.68 | **65.89** | **42.18** | **21.58** | 84.23 | 65.44 | 47.81 |
| **Mistral-7B** | Direct | 60.60 | 30.23 | 16.36 | 69.83 | 39.85 | 25.05 | 84.26 | 53.63 | 44.52 |
| | TAPE | 61.82 | NA | 6.13 | 58.92 | NA | 3.29 | 76.40 | NA | 16.44 |
| | GraphToken | 64.42 | 32.04 | 18.60 | 72.31 | 42.60 | 30.31 | 86.82 | 57.06 | **53.99** |
| | GraphSAGE | **68.12** | 43.09 | 25.77 | 75.51 | 52.94 | **34.29** | 87.51 | 66.57 | 52.74 |
| | GraphSAGE$_{co\text{-}train}$ | 67.61 | **43.14** | **27.20** | **76.96** | **55.46** | 33.26 | **87.61** | **66.75** | 52.97 |
| **CodeLlama-13B** | Direct | 57.55 | 28.88 | 14.29 | 68.57 | 41.79 | 24.10 | 91.20 | 76.07 | 66.40 |
| | TAPE | 64.03 | NA | 8.05 | 58.27 | NA | 2.01 | 77.74 | NA | 17.37 |
| | GraphToken | 62.15 | 32.55 | 20.08 | 74.57 | 47.60 | 35.06 | 92.50 | 73.57 | 69.42 |
| | GraphSAGE | 67.30 | 42.41 | 26.56 | 74.93 | 54.52 | **38.55** | **93.84** | **80.38** | 73.60 |
| | GraphSAGE$_{co\text{-}train}$ | **68.92** | **44.85** | **29.58** | **76.28** | **55.41** | 37.75 | 93.30 | 79.51 | **74.00** |
| **GPT-3.5-turbo** | Direct | 73.85 | 45.73 | 28.95 | 82.85 | 62.07 | 47.96 | 96.09 | 83.65 | 81.30 |
| | TAPE | 68.00 | NA | 8.83 | 62.43 | NA | 3.87 | 70.67 | NA | 8.92 |
| | GraphSAGE | **77.90** | 52.68 | 35.11 | 85.29 | 65.80 | 53.55 | **96.43** | **86.26** | **83.13** |
| | GraphSAGE$_{co\text{-}train}$ | 77.87 | **53.04** | **35.32** | **85.51** | **66.56** | **55.91** | 96.34 | 86.09 | 83.13 |
| **GPT-4-turbo** | Direct | 77.60 | 52.18 | 33.68 | 88.29 | 69.38 | 60.56 | 97.36 | 84.58 | 86.77 |
| | TAPE | 68.82 | NA | 9.50 | 63.94 | NA | 4.02 | 71.51 | NA | 9.40 |
| | GraphSAGE | **78.76** | 52.53 | **34.09** | 88.63 | 69.65 | 60.36 | 97.34 | 85.67 | **86.97** |
| | GraphSAGE$_{co\text{-}train}$ | 78.49 | **52.62** | 33.88 | **88.86** | **70.25** | **62.37** | **97.42** | **85.80** | 86.57 |

Table 3: **Performance Comparison of SGC and GraphSAGE on the UltraTool Benchmark [20]**. Integrating GNNs can lead to more significant improvements in LLMs on larger task graphs.

| LLM | Method | 0-shot | | | | 1-shot | | | |
|---|---|---|---|---|---|---|---|---|---|
| | | n-F1 ↑ | l-F1 ↑ | Acc ↑ | # Tok ↓ | n-F1 ↑ | l-F1 ↑ | Acc ↑ | # Tok ↓ |
| **CodeLlama-13B** | Direct | 38.88 | 16.42 | 13.58 | 10,535 | 57.64 | 30.44 | 26.25 | 10,737 |
| | BeamSearch | 49.71 | 22.51 | 17.08 | 26,008 | 64.93 | 36.23 | 33.47 | 23,023 |
| | SGC | 61.07 | 37.61 | 25.31 | 10,456 | 71.64 | 44.00 | 39.68 | 10,658 |
| | GraphSAGE | **63.78** | **39.91** | **27.98** | **10,456** | **72.81** | **45.26** | **43.49** | 10,658 |
| **GPT-3.5-turbo** | Direct | 54.35 | 21.35 | 18.33 | 8,462 | 63.58 | 30.85 | 25.00 | 8,614 |
| | BeamSearch | 55.40 | 28.02 | 19.76 | 21,979 | 63.41 | 34.05 | 26.28 | 20,813 |
| | SGC | 59.80 | 37.82 | 25.87 | 8,352 | 64.96 | 37.96 | 29.70 | 8,504 |
| | GraphSAGE | **63.97** | **42.26** | **30.35** | **8,352** | **70.49** | **47.79** | **39.74** | **8,504** |
| **GPT-4-turbo** | Direct | 68.63 | 40.01 | 27.20 | 8,513 | 69.54 | 41.79 | 28.17 | 8,693 |
| | BeamSearch | **71.29** | **43.99** | 30.40 | 18,793 | **71.99** | 44.54 | 31.62 | 20,515 |
| | SGC | 70.87 | 44.01 | 31.60 | 8,346 | 70.46 | 44.82 | 33.00 | 8,504 |
| | GraphSAGE | 70.67 | 43.83 | **34.40** | **8,346** | 70.75 | **47.68** | **37.22** | **8,504** |

training duration extends to only 15 minutes when GNNs are jointly trained with e5-335M model. This efficiency stands in stark contrast to the 10-20 hours required for tuning open-sourced LLMs.

## 5.4 Scaling to Large Task Graphs

To demonstrate the scalability of our method to larger task graphs, we conducted a supplementary experiment on the newly released planning benchmark **UltraTool** [20], which features a relatively large task graph with 260 nodes. Details on processing this dataset are provided in Appendix C.3. We present a performance comparison of GNN models (training-free SGC and training-required GraphSAGE) against strong baselines like BeamSearch in Table 3. Among the metrics, accuracy (*Acc*) is calculated based on whether the predicted tasks match the ground-truth tasks, measuring the success rate at each case level. In such conditions, integrating a GNN significantly enhances performance and mitigates planning failures, e.g., GPT-4-turbo undergoes a **9.05% accuracy improvement** with the introduction of GraphSAGE.

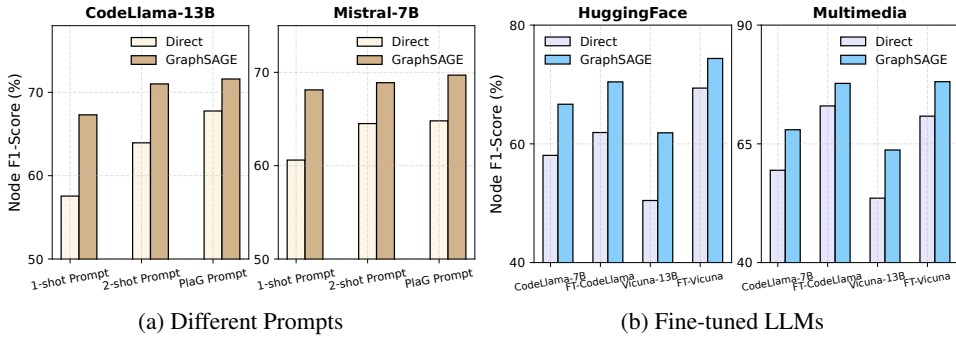

(a) Different Prompts        (b) Fine-tuned LLMs

Figure 3: **Orthogonal Effectiveness to both Improved Prompts and Fine-tuned LLMs**

The results indicate that (1) LLMs' performance is vulnerable to the scale of task graphs; (2) The performance gain of the proposed method increases with a larger task graph.

### 5.5 Improved Prompts and Fine-tuned LLMs

In this subsection, we show that the proposed method is orthogonal to two dominant methods, i.e., prompt engineering and fine-tuning.

**Orthogonal to Improved Prompts:** We investigate GNN's effectiveness when applied to improved prompt templates, i.e., strategically designed prompts that enhance the task planning abilities of LLMs. Specifically, we consider two types of prompts: **(1) In-context Learning with Increased Examples [45]** During main experiments, we maintain the consistent 1-shot in-context learning example for LLM's direct inference. To realize further improvements, we increase the number of examples to 2, and results under this setting are denoted as "2-shot Prompt"; **(2) Plan like a Graph (PlaG) [30]** We adopt the prompt in [30] to encourage LLM to think and plan in a graph-like manner. Specifically, we convert the entire task graph into plain text and then integrate PlaG instructions to enhance LLM's planning. Results under this prompt template are denoted as "PlaG Prompt".

From the results shown in Figure 3a, where we apply three different prompts to CodeLlama-13B and Mistral-7B on HuggingFace, it is clear that applying GraphSAGE to improved prompts, where task steps are more concisely decomposed and predictions are more accurate, can also boost performance.

**Orthogonal to LLMs' Fine-tuning:** To explore whether our framework maintains effectiveness on fine-tuned LLMs, which have acquired dataset-specific task planning capabilities, we conduct further experiments. For each dataset, we use LoRA [19] to fine-tune two LLMs of different parameter scales, including CodeLlama-7B and Vicuna-13B, based on the same training data as GNNs. Details of fine-tuning process are provided in Appendix F.3. The finetuned model is named as "FT-CodeLlama" and "FT-Vicuna" in Figure 3b.

The results depicted in Figure 3b demonstrate that fine-tuning markedly enhances the task-planning capabilities of LLMs. Furthermore, applying GraphSAGE to the decomposed tasks of LLMs further improves the accuracy of task planning.

## 6 Conclusions

This paper presents an initial exploration into graph-learning-based approaches for task planning in language agents. Through theoretical analysis, we demonstrate the inductive bias of the attention mechanism and the utility of auto-regressive loss impedes their effectiveness in task planning. We propose to integrate GNNs for task graph analysis, which yields performance improvements across a range of LLMs and planning benchmarks.

**Limitations:** Despite the encouraging performance, there are limitations that highlight significant opportunities for enhancement. Firstly, our proposed method, while effective, is straightforward; more sophisticated graph-learning-based decision-making algorithms could potentially offer further improvements. Secondly, the construction of the task graph currently requires manual effort. Investigating automated graph generation techniques for diverse applications is another promising direction for future work.

## Acknowledgements

This work is partly supported by grant from the Research Grants Council of the Hong Kong Special Administrative Region, China (No. CUHK 14217622).

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

# Contents

# A   Related Works and Discussions

## A.1   Planning Algorithms in LLMs

The existing studies of task planning approaches can be categorized into several directions, including task decomposition, multi-plan selection, the use of external planners, reflection, and memory-aided planning [22]. Task decomposition methods, such as the chain-of-thought approach [63], employ the divide-and-conquer strategy, utilizing LLMs for both task decomposition and sub-task planning. The application of this method to task planning is detailed in Section 2.2 and is referred to as "Direct" in the baseline comparison. Multi-plan selection strategies, exemplified by the tree-of-thought [71] and graph-of-thought [4], leverage search-based methods to generate plans. Subsequently, LLMs evaluate these plans to select the most effective one. The "GraphSearch" methods used in our baselines fall into this category. External planner approaches [31] use LLMs to convert the problem into Planning Domain Definition Language (PDDL) and then employ classic solvers to address the planning problem. PDDL requires a pre-defined goal, for example, moving the blocks from one state to another state. However, the goal of task planning investigated in language agents deals with more flexible and personal goals, spanning personal needs in video, text, and image processing. Translating these goals into formal PDDL is very difficult. We instead demonstrate that GNNs can serve as an effective external planner in this application. Reflection-based methods [48] focus on reflecting upon experiences to refine the plan, while memory-aided planning approaches [73] utilize external experiences, such as those from search engines. These approaches are deployed in interactive environments and orthogonal to this paper.

## A.2   Task Planning in Traditional AI

Apart from task planning in language agents, there is also a domain in traditional AI called task planning [1, 6, 18, 55, 44, 51, 35, 10]. Task planning in traditional AI is defined as $(\mathcal{S}, \mathcal{A}, \mathcal{T}, \mathcal{C}, \mathcal{G}, s_0)$, where $\mathcal{S}$ is the states, $\mathcal{A}$ is the action space, $\mathcal{T} : \mathcal{S} \times \mathcal{S} \times \mathcal{A} \to [0, 1]$, a cost function $\mathcal{C} : \mathcal{S} \times \mathcal{A} \to [0, \infty)$, a set of goal states $\mathcal{G} \subseteq \mathcal{S}$, and an initial state $s_0$. An agent following a policy $\pi : \mathcal{A} \times \mathcal{S} \to [0, 1]$ will start in state $s_0$, then repeatedly choose an action $a \sim \pi(a|s)$ and execute it to reach a new state $s' \sim \mathcal{T}(s'|s, a)$, incurring a cost $\mathcal{C}(s, a)$ along its way. An optimal policy $\pi^*(a|s)$ is one that reaches the goal state with probability $1$ while minimizing the total expected cost. Traditionally, task planning is solved by reinforcement learning approaches [6] and heuristic A* approaches [18]. The neural network-based approaches are employed to accelerate the computation and improve the performance [55, 44, 51, 35, 10].

The task planning in language agents is a different application. It cannot be solved as a constraint satisfaction problem, since both the features of task graph and user request are expressed in natural language. Compared with traditional planning, task planning for language agents involves diverse and open-ended goals due to the varied personal requirements users have. For example, on platforms like Hugging Face, users' intentions span across video, text, and image domains. On the contrary, classic planning has a fixed goal for a given domain, e.g., in the n-puzzle [55], the goal is formally as placing the tiles in numerical order. Within this new application domain, while existing research primarily focuses on prompt design for pre-trained LLMs, our work underscores the importance of traditional planning methods, such as GNNs, in complementing LLMs.

## A.3   Planning in Agents and Neuroscience

Planning is a pivotal topic in both agents and neuroscience, where graphs play an indispensable role. We believe the concepts and insights presented in this paper are useful to these fields.

The TaskBench, RestBench, and UltraTool dataset used in this paper belongs to the tool agents. The Math agent, AlphaGeometry, employs LLMs to generate auxiliary constructions in geometry [56]. Considering lemmas as nodes and their interdependencies as edges, the endeavor to prove a theorem resembles the task of identifying a route to the theorem node within the graph constituted by potential lemma nodes and the edges that represent their interdependencies. There are no explicit task graphs in game agents [58], embodied agents [21], and code agents [48]. The core strategy in these domains is to employ verbal reinforcement learning within LLMs. The state and transitions in reinforcement learning can be modeled as nodes and edges in the graph. In addition, there are case-by-case graph models in these agent applications. For example, in code agents, one can view the class as the nodes

and dependencies as the edges. In the embodied agents, the objects in the environment can be viewed as the nodes.

In the neuroscience, animal planning is often assessed through path planning in mazes [2, 64]. Inspired by these animal experiments, planning testbenches have been developed for LLMs [37]. A computational model known as the Tolman-Eichenbaum Machine (TEM) has been proposed to decipher the mechanisms of general planning in animals across various environments, such as mazes [64]. The TEM model posits that hippocampal cells, including place and landmark cells, remap between environments, while entorhinal cells exhibit a range of properties that mirror spatial responses, including grid, band, border, and object-vector cells. In essence, hippocampal cells map sensory inputs onto locations in abstract graphs and remap, and entorhinal cells execute graph operations.

## A.4 LLMs for Graphs

With the breakthroughs in LLMs, there has been a surge of interest in applying LLMs to graph-related problems [27, 28]. GPT4Graph [14] and NLGraph [59] are two prominent benchmarks designed to evaluate the performance of LLMs in the context of graph tasks. They encompass a wide spectrum of challenges, various input formats, and state-of-the-art prompting techniques, demonstrating that LLMs possess basic graph processing capabilities. Importantly, the choice of prompts and formats significantly influences performance. However, these benchmarks also expose the models' susceptibility to spurious correlations within graphs. For instance, GPT-4 achieves only about $50\%$ accuracy on shortest-path tasks, even with the use of complex prompts. GraphInstruct [34] attempts to fine-tune LLMs on graph-theory-related tasks, resulting in improved performance, though it remains far from satisfactory. Despite these empirical efforts, there is a limited theoretical understanding of these evaluation results. The analysis in Section 3.3 aims to shed light on the empirical observations reported in these studies.

Considering these negative results, a new line of research has emerged that utilizes the output of GNNs as tokens for LLMs, as seen in GraphGPT [53], GraphLLM [9], GraphToken [38], and G-Retriever [17]. These approaches have demonstrated significant improvements in performance on GNN-related tasks. However, they have not yet been applied to task planning due to the lack of extensive training data. A promising future direction involves using task planning data generated by GPT to fine-tune graph foundation models, such as GraphGPT [53], and applying them to task planning. This paper proposes to use task planning as a new benchmark for this line of research.

## A.5 Theoretical Analysis of Reasoning

Reasoning is closely related to task planning and decision-making. The theoretical exploration of the reasoning abilities of neural networks was initiated by [68]. This work unifies various reasoning tasks, such as intuitive physics, visual question answering, and shortest path calculations, into DP problems. It then analyzes the generalization capabilities of MLPs, DeepSets, and GNNs. It is demonstrated that GNNs exhibit the best generalization bounds, attributed to their architecture's resemblance to the Bellman-Ford algorithm, which is adept at solving DP problems. In terms of reasoning abilities within LLMs, [12] examines how the Chain of Thought (CoT) approach aids in solving arithmetic and DP problems without graphs. By decomposing challenging problems into simpler subproblems, CoT extends the expressive capabilities of Transformers from $TC^0$ to P. This analysis is further applied to linear and sparse Transformers in [70].

Our proof of Theorem 1 builds upon the proof of Theorem 4.7 in [12]. However, while [12] addresses DP problems without graph structures, Theorem 1 specifically focuses on DP problems with graph edge list inputs. Moreover, unlike [12], which decomposes and solves the DP problem sequentially, Theorem 1 proposes a method to simulate DP on edge lists in parallel. In addition, we analyze the negative results rising from the inductive bias of attention mechanism and auto-regressive loss. These theoretical contributions are novel and promise to be valuable for general reasoning and planning tasks.

## A.6   GNNs and GraphSearch for Combinatorial Optimization

GNNs are popular approaches for solving decision-making problems on graphs. The problems investigated are typically NP-hard, such as the minimum vertex cover, maximum cut, and the traveling salesman problem [8]. The basic approach involves selecting nodes one by one in a manner that satisfies the constraints [24]. In this paper, we adopt this method to sequentially select task nodes. Furthermore, reinforcement learning can be used to enhance the performance of GNNs beyond what is achievable with supervised labels alone [24]. In [24], the node with the highest score is selected exclusively. Conversely, [29] employs beam search to improve performance by selecting the top-$k$ nodes in a single iteration. Additionally, GNNs are utilized as the method for variable selection in exhaustive searches for exact solutions to combinatorial optimization problems [13]. This paper conceptualizes task planning as a graph-based decision-making problem. Both greedy and beam search algorithms have been adopted in task planning [33, 32]. Given this connection, a promising future direction involves repurposing GNNs for decision-making approaches in task-planning applications.

# B Prompts

Table 4: **Prompt template for LLM's direct inference [45]**

---

\# TASK LIST \#
```
{{ task list }}
```

\# GOAL \#
Based on the above tasks, I want you to generate task steps and a task invocation graph (including nodes and edges) to address the \# USER REQUEST \#. The format must be in strict JSON format, like:
{
  "task_steps": [ step description for one or more steps ],
  "task_nodes": [{
      "task": " task name must be from \# TASK LIST \# ",
      "arguments": [ a concise list of arguments for the task ]
  }],
  "task_links": [{ "source": "task name i", "target": "task name j" }],
}

\# REQUIREMENTS \#
1. Generated task steps and task nodes can resolve the user request \# USER REQUEST \# perfectly. Task name must be selected from \# TASK LIST \#.
2. The task steps should strictly align with the task nodes, and the number of task steps should be same with the task nodes.
3. The task links should reflect the temporal and resource dependencies among task nodes, i.e., the order in which the tasks are invoked.

\# EXAMPLE \#
```
{{ in-context learning examples }}
```

\# USER REQUEST \#
```
{{ user request }}
```

Now, please generate your response in a strict JSON format: \# RESULT \#

---

Table 5: **Prompt templates of GraphSearch [33]**

| Scenario | Prompt |
|---|---|
| **Task Assessment** | # CANDIDATE TASK LIST #
`{{ candidate tasks }}`

# GOAL #
Based on the provided # CANDIDATE TASK LIST # and the user's request described in the # STEP #, generate a score dictionary to assess each task's problem-solving abilities. The output must be in a strict JSON format, like: { "candidate task name 1": score, ... }.

# REQUIREMENTS #
1. The keys of the generated score dictionary must align with the provided candidate tasks, and you should output scores for all candidate tasks.
2. The "score" field denotes a concrete score that assesses whether each task can solve the given step's demand. The score should be in the range of $[1, 2, 3, 4, 5]$, where a higher score indicates better task-solving and matching abilities.
3. Carefully consider the user's intention in # STEP # to assign the score. If the # STEP # contains a candidate task, its score should be >= 3.

# EXAMPLE #
`{{ in-context learning examples }}`

# STEP #
`{{ step description }}`

Now please generate your result in a strict JSON format: # RESULT # |
| **Path Selection** | # GOAL #
Based on the provided # USER REQUEST # and initially inferred # STEPS #, select the best path solution list from # SOLUTION LIST #. The selected solution should be the one that can perfectly solve the user's request and strictly align with the inferred steps. The output must be in strict JSON format, like: { "best_solution": [list of invoked tasks ]}

# REQUIREMENTS #
1. Carefully analyze both the user's request and previously inferred task steps. Select the best solution that can perfectly follow the inferred steps and solve user's request. Do not change their corresponding sequences.
2. Make sure that each task in the final solution list exists in the valid # TASK LIST # `{{ task list }}`.

# USER REQUEST #
`{{ user request }}`

# STEPS #
`{{ steps }}`

# SOLUTION LIST #
`{{ list of searched solutions }}`

Now please generate your result in a strict JSON format: # RESULT # |

Table 6: **Prompt template for LLM's filling in invocation parameters [45]**

# GOAL #
Given a # USER REQUEST # and # PLANNED TASKS # to be invoked in sequence to solve this request, please fill up each invoked task's invocation parameters.
The format must be in strict JSON format, like:
{
  "task_nodes": [{
    "task": " task name must be from # PLANNED TASKS # ",
    "arguments": [ a concise list of arguments for this task ]
  },...]
}

# REQUIREMENTS #
1. Consider each task's input and output requirements, and carefully fill in the arguments for each task.
2. Analyze the resource dependencies, keeping in mind that these tasks are invoked sequentially to address the original request.
3. The number of predicted task_nodes must strictly align with the provided tasks.

# USER REQUEST #
{{ user request }}

# PLANNED TASKS #
{{ a list of previously GNN retrieved tasks }}

# DETAILS OF TASKS #
{{ details, i.e., input and output requirements of each planned task }}

# EXAMPLE #
{{ in-context learning examples }}

Now, please generate your response in a strict JSON format: # RESULT #

Table 7: **Statistics of Experimental Datasets**

| Type | Statistic | TaskBench | | | RestBench TMDB | UltraTool |
|------|-----------|-----------|---|---|---------|-----------|
| | | HuggingFace | Multimedia | Daily Life | | |
| **Task Graph** | # Node | 23 | 40 | 40 | 46 | 260 |
| | # Links | 225 | 449 | 1560 | 979 | 611 |
| | Link Type | Resource | Resource | Temporal | Resource / Category | Resource |
| **All Data** | # Samples | 7,546 | 5,584 | 4,320 | 100 | 3,527 |
| **Test Set** | # Samples | 500 | 500 | 500 | 94 | 500 |
| | # Avg Nodes | 3.81 | 3.92 | 4.05 | 2.33 | 2.38 |
| | # Avg Links | 2.81 | 2.92 | 3.05 | 1.33 | 1.38 |

# C  Datasets

## C.1  Overview

We provide the statistics of experimental datasets from three task planning benchmarks in Table 7. The illustrative examples from each dataset are shown in Figure 4. For datasets from TaskBench [45], each sample consists of original user request, corresponding decomposed task steps, and ground-truth task invocation path. As RestBench [50] and UltraTool [20] include only user requests and corresponding API invocation sequences, we prompt GPT-4 to infer decomposed task steps aligned with each invoked API, thereby finalizing the dataset.

## C.2  Reformatting Details of RestBench

The TMDB dataset from RestBench, focuses on movie-related searching and recommending functions. To align RestBench with our experiments, we have implemented the following processing steps:

**Reformatting original APIs by assigning unique task names and descriptions:** APIs in RestBench were represented by request paths, such as "`GET /movie/top_rated`", referring to the API that retrieves top-rated movies on TMDB. To enhance semantic differentiation among APIs, we first prompt GPT-4 to assign a unique name and a detailed functional description to each API. These names and descriptions were then manually verified and refined. For example, the API previously mentioned is renamed "`Get Top-Rated Movies`" with the description: "This API retrieves a list of the highest-rated movies." Note that though the original TMDB dataset contains $54$ APIs, some were never invoked in any data examples. Therefore, we focus only on those APIs that appeared in at least one user request, resulting in a refined set of $46$ APIs.

**Constructing a Task Graph:** Each API is regarded as a unique task node, and we depict their relationships from two aspects: (1) categorical association, and (2) resource dependencies. For instance, APIs that provide movie-related functionalities, such as retrieving movie details or recommending films, are grouped under the *movie* category. Conversely, APIs focused on person-related functions, like searching for actors, are classified under the *person* category. Additionally, if two APIs share a common parameter like `move_id`, we establish a link between them to indicate a resource dependency.

**Reformatting Raw Data Examples:** The original data samples in RestBench included a single query and its corresponding API invocation sequence. To reformat this data into a path structure, we treated each invoked API as a node and the sequence of invocations as directed links from one API to the next. For example, a TMDB data sample consists of the query "*Who was the lead actor in the movie The Dark Knight*" and corresponding ground-truth API solution ["`GET /search/movie`", "`GET /movie/{movie_id}/credits`"]. We transform this solution into a task invocation path as "{ "task_nodes": ["`Search Movie`", "`Get Movie Credit`"], "task_links": [{"source": "`Search Movie`", "target": "`Get Movie Credit`"}]} ".

## C.3  Reformatting Details of UltraTool

**Motivation:** In our main experiments using TaskBench [45] and RestBench [50], we observed that strong LLMs like GPT-4-turbo already perform well. This may be attributed to two factors: (1) the training of GPT-4-turbo likely included knowledge relevent to these benchmarks, as both were released before GPT-4 and utilize popular platforms such as HuggingFace and TMDB; (2) the task

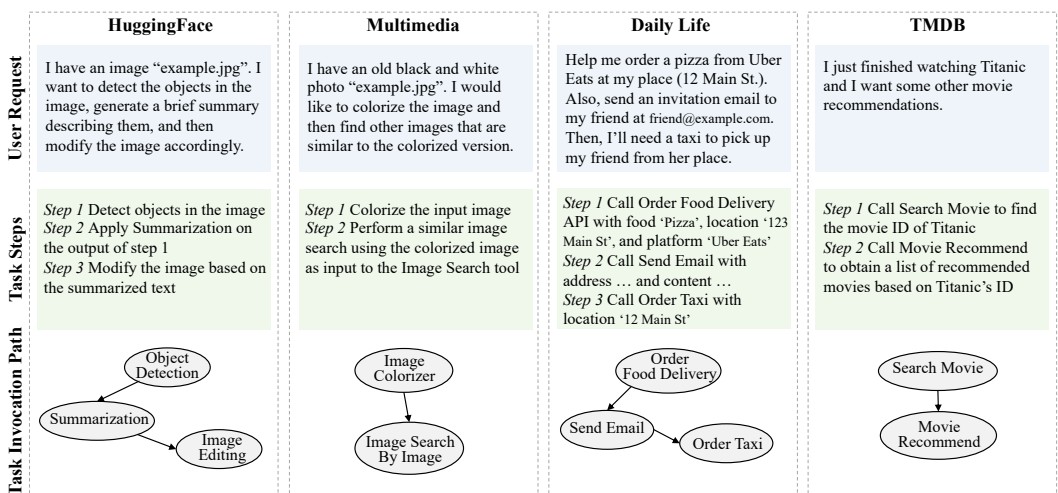

Figure 4: **Illustrative details of experimental datasets.**

graphs are relatively small, containing no more than 50 tasks, which falls within the capabilities of LLMs. Therefore, we aim to evacuate planning performance as well as GNN's effectiveness in a more challenging scenario, i.e., one that requires specific knowledge beyond the LLM's training and involves larger task graphs that possibly exceed its memory and reasoning capabilities.

**Constructing a Task Graph:** We utilized UltraTool [20], released in March 2024, which features complex planning scenarios across thousands of tasks, including travel, tourism, and other daily life domains. The original UltraTool contains 5, 824 samples with 2, 032 available tasks across 22 distinct domains. However, we observed that some tasks appear in only one sample, making them quite rare. Therefore, we filtered this dataset as follows: first, we considered only tasks that appeared more than 5 times across all samples, focusing on more common tasks that cater to daily life. Next, we retained samples that incorporated these filtered tasks, ensuring that the number of invoked tasks exceeded 2 to satisfy the multi-task planning scenario. After this filtering, we identified 260 **distinct tasks**. We then constructed the task graph by treating each task as a node and adding links between tasks that were invoked sequentially in the dataset. For each task, we further verified and refined its functional descriptions to ensure semantic suitability.

**Reformatting Raw Data Examples:** We allocated 500 samples for testing and 3, 000 samples for GNN training. Although the original UltraTool provided decomposed steps, we found them too coarse-grained, making it difficult to align step descriptions with suitable tasks. Therefore, we employed a similar strategy by prompting GPT-4 to infer decomposed task steps that align with each invoked task. As a result, each data sample consists of the user request, corresponding steps, and ground-truth invoked tasks, ensuring high quality for GNN training.

# D   Supplementary Materials for Theoretical Results

## D.1   Dynamic Programming

**Longest Increasing Subsequence:** The Longest Increasing Subsequence (LIS) problem is a classic dynamic programming problem that involves finding the length of the longest subsequence within a given array `arr` where the elements are in strictly increasing order. The state transition function for the LIS problem can be expressed as:

$$\text{Answer}[k][i] = \max_{j \in \mathcal{T}(i)} (\text{Answer}[k-1][j] + (\mathbb{I}(j \neq i) \times 1)),$$

where $\mathcal{T}(i) = \{i\} \cup \{j \,|\, j < i \text{ and } \texttt{arr}[j] < \texttt{arr}[i]\}$ denotes the set of states that can transfer to state $i$, the aggregation function $g(x,y)$ is implemented as $\max(x,y)$, and the cost $c[i][j]$ is 1 for those candidate states that are not equal to state $i$ as adding the element leads to a longer subsequence while 0 for the state itself.

**Bellman-Ford Algorithm:** The Bellman-Ford algorithm is also a classic dynamic programming algorithm used to find the shortest paths from a single source vertex to all other vertices in a weighted graph. The core idea behind the Bellman-Ford algorithm is that the distance from the source vertex to a target vertex can be computed as the minimum distance from the source to any of the target's neighboring vertices, plus the weight of the edge connecting the neighbor to the target. Therefore, the state transition function for the Bellman-Ford algorithm can be expressed as:

$$\text{Answer}[k][i] = \min_{j \in \mathcal{T}(i)} (\text{Answer}[k-1][j] + w[j][i]),$$

where $\text{Answer}[k][i]$ represents the length of the shortest path from the source vertex to node $i$ at the $k$-th iteration, $\mathcal{T}(i) = \mathcal{N}^-(i)$ denotes the set of in-neighbors of node $i$, and $w[j][i]$ denotes the weight from node $j$ to $i$. The aggregation function $g(x,y)$ is implemented as $\min(x,y)$ as we try to find the shortest path.

**Travelling Salesman Problem (TSP):** This problem is defined as, given a set of cities and the distances between every pair of cities, finding the shortest possible route that visits every city exactly once and returns to the starting point. If we regard the set of already visited city $\mathcal{S}$ ending at $i$-th city as the current state, then states that can transfer to current state are those that ending city can reach $k$. Therefore, the state transition function for the TSP problem can be expressed as:

$$\text{Answer}[k][i][\mathcal{S}] = \min_{j \in \mathcal{S}, j \neq i} (\text{Answer}[k-1][j][\mathcal{S} \setminus \{i\}] + w[j][i]),$$

where $\text{Answer}[k][i][\mathcal{S}]$ represents the cost of the shortest tour that visits all the cities in the set $\mathcal{S}$ and ends at the $i$-th city, the aggregation function $g(x,y)$ is still implemented as $\min(x,y)$ since we aim to find the shortest path, and $w[j][i]$ denotes the distance from city $j$ to city $i$.

## D.2   Proof of Theorem 1

**Assumption 1.** Each function $f, g$ in (2) can be approximated by constant size MLP.

**Assumption 2.** The aggregation function $\square$ in (2) is one of min, max, sum, mean.

The first assumption is mild as MLPs are universal approximators. The second assumption is mild because these are the most commonly used aggregation functions.

**Theorem 3.** *(Expressiveness) Assume the input format is given in* (1) *and* $f, g, \square$ *in DP update* (2) *satisfy the assumptions 1 and 2. There exists a log-precision constant-depth and constant-width Transformer that simulates* 1 *steps of DP update in* (2)*. As a consequence, there exists a log-precision* $O(k)$*-depth and constant-width Transformer that simulates* $k$ *steps of DP update in* (2)*.*

*Proof.* **Token Embedding and Positional Embedding:** The three-dimensional token embedding includes the token type $e^{\text{type1}}$ (0 for answer; 1 for node; 2 for edge cost), refined token type $e^{\text{type2}}$ (0 for answer, 1 for the node tokens in initial states, 2 for the target node tokens in the edge list, 3 for the source node tokens in the edge list, 4 for edge cost), and the token id $e^{\text{token}}$ (from 0 to $|V| - 1$). The two-dimensional positional embedding includes the embedding for initial state tokens $e^{\text{pos1}}$ (0 for edge list tokens, 1 for the first two elements of initial states, 2 for the second two elements

of initial states, etc.), embedding for edge list tokens $e^{\text{pos2}}$ (0 for initial state tokens, 1 for the first three elements of the edge list, 2 for the second three elements of the edge list, etc.). There are also constant-dimensional placeholders to put the states of DP.

**Block 1 - Initial State Broadcast:** The goal of the first block is to broadcast the initial states from the initial state token to node tokens. (1) Use MLPs to recover the digits of the answer tokens and put them in the first placeholder if $e_k^{type} == 0$; (2) Copy the first placeholder from answer token to its previous node token by using **COPY** in Lemma 1 and setting $\mathcal{S}_k = \{j|(e_k^{\text{pos1}} - e_j^{\text{pos1}})^2 < \delta\}$; (3) Broadcast the first placeholder with **SUM** in Lemma 1 and setting $\mathcal{S}_k = \{j|(e_k^{\text{type1}} - e_j^{\text{type1}})^2 + (e_k^{\text{token}} - e_j^{\text{token}})^2 < \delta\}$. Now the state for every node token $u_i$ is $[e^{\text{type1}}, e^{\text{type2}}, e^{\text{token}}, e^{\text{pos1}}, e^{\text{pos2}}, \text{Answer}[0][u_i]]$.

**Block 2 - Edge Feature Operations:** The goal of the second block is to copy the edge features from the edge feature token to the corresponding node tokens. (1) Use MLPs to recover the digits of the edge feature tokens and put them in the second placeholder if $e^{\text{type1}} == 2$; (2) Copy the second placeholder from the edge feature token to the node token by using **SUM** in Lemma 1 and setting $\mathcal{S}_k = \{j|(e_k^{\text{pos2}} - e_j^{\text{pos2}})^2 < \delta\}$. Now the state for every node token $u_i$ of the $i$-th edge is $[e^{\text{type1}}, e^{\text{type2}}, e^{\text{token}}, e^{\text{pos1}}, e^{\text{pos2}}, \text{Answer}[0][u_i], c[u_i][v_i]]$.

**Block 3 - Message Preparation:** (1) Use MLPs to compute $g$ and place the results in the third placeholder. Now the state for every node token $u_i$ of the $i$-th edge is $[e^{\text{type1}}, e^{\text{type2}}, e^{\text{token}}, e^{\text{pos1}}, e^{\text{pos2}}, \text{Answer}[0][u_i], c[u_i][v_i], g(\text{Answer}[0][u_i], c[u_i][v_i])]$; (2) Use MLPs to clean up the first and second placeholder. Now the state for every node token $u_i$ is $[e^{\text{type1}}, e^{\text{type2}}, e^{\text{token}}, e^{\text{pos1}}, e^{\text{pos2}}, g(\text{Answer}[0][u_i], c[u_i][v_i])]$

**Block 4 - Message Passing:** The goal of the fourth block is to compute $\square$ and $f$. (1) Use one or two attention heads (one for max, min, mean aggregations, and two for sum aggregations) to perform the aggregation operation. This is achieved by using **MEAN** or **MAX** or **SUM** for the first placeholder and setting $\mathcal{S}_k = \{j|(e_k^{\text{token}} - e_j^{\text{token}})^2 + (e_k^{\text{type2}} - e_j^{\text{type2}})^2 < \delta\}$. Now the state for every node token $u_i$ is $[e^{\text{type1}}, e^{\text{type2}}, e^{\text{token}}, e^{\text{pos1}}, e^{\text{pos2}}, \square_{v_j \in \mathcal{T}(u_i)} g(\text{Answer}[0][u_i], c[u_i][v_j])]$; (2) Use MLPs to compute $f$. Now the state for every node token is $[e^{\text{type1}}, e^{\text{type2}}, e^{\text{token}}, e^{\text{pos1}}, e^{\text{pos2}}, f(\square_{v_j \in \mathcal{T}(i)} g(\text{Answer}[0][u_i], c[u_i][v_j]))]$.

After four blocks, the final state for every node token $u_i$ is given by $[e^{\text{type1}}, e^{\text{type2}}, e^{\text{token}}, e^{\text{pos1}}, e^{\text{pos2}}, \text{Answer}[1][u_i]]$. $\text{Answer}[k][u_i]$ can be obtained by repeating the above four blocks $k$ times.

$\square$

**Lemma 1.** *[12] Let $n \in \mathbb{N}$ be an integer and $\boldsymbol{x}_1, \cdots, \boldsymbol{x}_n$ be a sequence of vectors where $\boldsymbol{x}_i = (\tilde{\boldsymbol{x}}_i, r_i, 1) \in [-M, M]^{d+2}$ where $M$ is a large constant. Let $\boldsymbol{K}, \boldsymbol{Q}, \boldsymbol{V} \in \mathbb{R}^{d' \times (d+2)}$ be any matrices with $\|\boldsymbol{V}\|_\infty \leq 1$ and let $0 < \rho, \delta < M$ be any real numbers. Denote $\boldsymbol{q}_i = \boldsymbol{Q}\boldsymbol{x}_i, \boldsymbol{k}_j = \boldsymbol{K}\boldsymbol{x}_i, \boldsymbol{v}_j = \boldsymbol{V}\boldsymbol{x}_j$. Define a matching set $\mathcal{S} = \{j||\boldsymbol{q}_i^T \boldsymbol{k}_j| \leq \rho\}$. Define two following operations*

- *COPY: The output is a sequence of vectors $\boldsymbol{u}_1, \cdots, \boldsymbol{u}_n$ with $\boldsymbol{u}_i = \boldsymbol{v}_{pos(i)}$, where $pos(i) = \arg\max_{j \in \mathcal{S}_i} r_j$.*

- *MEAN, MAX, SUM: The output is a sequence of vectors $\boldsymbol{u}_1, \cdots, \boldsymbol{u}_n$, where $\boldsymbol{u}_i = \square_{j \in \mathcal{S}_i} \boldsymbol{v}_j$ and $\square$ is min or max or sum or mean.*

*Specifically, for any sequence of vectors $\boldsymbol{x}_1, \boldsymbol{x}_2, \cdots, \boldsymbol{x}_n$, denote the corresponding output of the attention layer as $\boldsymbol{o}_1, \boldsymbol{o}_2, \cdots, \boldsymbol{o}_n$. Then, we have $\|\boldsymbol{u}_i - \boldsymbol{o}_i\|_\infty \leq \epsilon$ for all $i \in [n]$ and $\mathcal{S} \neq \emptyset$.*

### D.3 Permutation Invariance Test of LLMs

We test whether LLMs respect the permutation invariance property in graph problems and the results are given in Figure 5.

### D.4 Proof of Proposition 1

**Proposition 2.** *Assume the input format is as described in Equation (1) and that the attention mechanism is limited to attending to a constant number of tokens. There exists at least one instance*

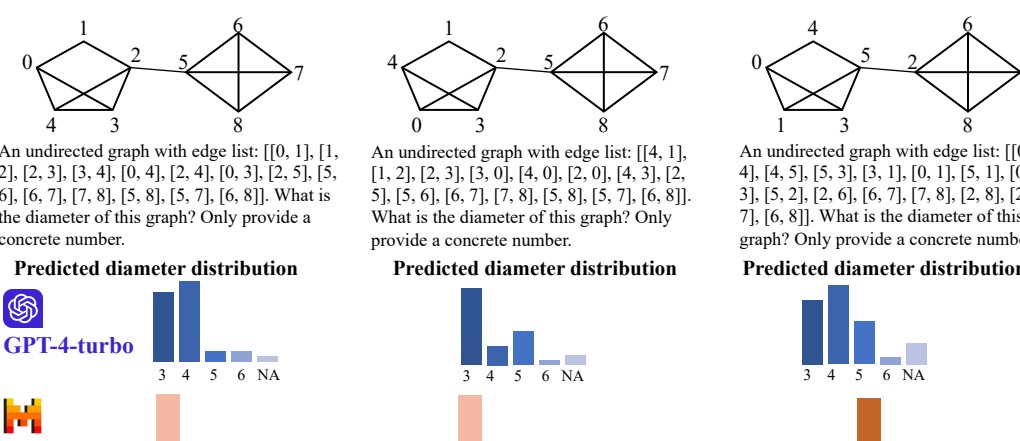

Figure 5: **Illustrative Examples of LLMs Failure to Solve Graph Computational Problems under Permutation (i.e., node re-odering)**. Experiments were conducted for **30** times.

*of one-step DP update such that no log-precision constant-width constant-depth transformer can simulate.*

*Proof.* We present a proof by contradiction. Assume that a token in a Transformer with a constant depth, constant width, and log-precision can attend to only a constant number of nodes. Under this assumption, the total information accessible to the token in such a Transformer architecture amounts to $O(\log n)$ bits. However, for a graph with $|V|$ nodes, the number of possible outcomes from executing one-step DP is $O(e^{|V|})$, necessitating $\Theta(|V|)$ bits for representation. By the pigeonhole principle, this scenario inevitably leads to at least two distinct DP outcomes being represented by the same output sequence generated by the model, thereby constituting a contradiction. $\square$

### D.5 Proof of Theorem 2

**Theorem 4.** *(Spurious correlations of auto-regressive loss) Assume (1) the loss employed is a next-token-prediction loss utilizing cross-entropy, applied to the sub-sequence $v_1 \, v_2 \, \cdots \, t$ during training; (2) the output logits are determined by target node $t$ and the current node $v_{i-1}$. Let $N_{t,v_{i-1},u}$ be the number of times in the training dataset such that $t$ is the target node, $v_{i-1}$ is the current node and $v_i = u$ is the next node. The optimal logits for predicting the next node $u$ from current node $v_{i-1}$ towards target node $t$ is given by $\hat{v}_i[u] = \frac{N_{t,v_{i-1},u}}{\sum_u N_{t,v_{i-1},u}}$ if $\sum_u N_{t,v_{i-1},u} > 0$. If $\sum_u N_{t,v_{i-1},u} = 0$, $\hat{v}_i[u]$ can be any non-negative number subject to $\sum_u \hat{v}_i[u] = 1$.*

*Proof.* We denote $\mathcal{D}$ as the training dataset, $L_i$ as the sequence length of the $i$-th sequence in the dataset, $v_{i,j}$ as the one hot embedding of the $j$-th token in the $i$-th training sequence, and $\hat{v}_{i,j,u}$ as the $u$-th logit at the $j$-th token in the $i$-th sequence. The cross-entropy loss is given by

$$- \sum_{i \in [|\mathcal{D}|]} \sum_{j=4}^{L_i} v_{i,j,u} \log \hat{v}_{i,j,u} = - \sum_{i \in [|\mathcal{D}|]} \sum_{j=4}^{L_i} \mathbb{I}_{u=v_{i,j}} \log \hat{v}_{i,j,u} \overset{(a)}{=} - \sum_{t,v_{j-1}} \sum_u N_{t,v_{j-1},u} \log \hat{v}_{i,j,u}$$

$$\overset{(b)}{=} - \sum_{t,v_{j-1},u} \left( \sum_u N_{t,v_{j-1},u} \right) \left[ \left( \frac{N_{t,v_{j-1},u}}{\sum_u N_{t,v_{j-1},u}} \right) \log \hat{v}_{i,j,u} \right],$$

where (a) uses the assumption that the output logits are determined by target node $t$ and the current node $v_{i-1}$. In (b), we assume that $\sum_u N_{t,v_{i-1},u} \neq 0$. The cross-entropy is minimized when $\hat{v}_{i,j,u} = \frac{N_{t,v_{j-1},u}}{\sum_u N_{t,v_{j-1},u}}$. If $\sum_u N_{t,v_{i-1},u} = 0$, then the corresponding logits will not affect the loss function and $\hat{v}_{i,j,u}$ can take any number. $\square$

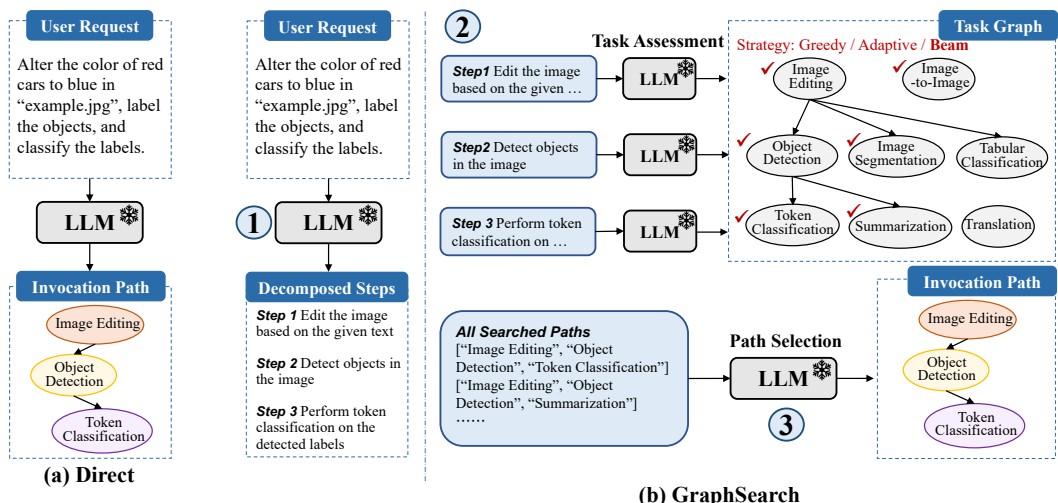

Figure 6: **Illustration of LLM's Direct Inference and GraphSearch Method.**

# E    Supplementary Materials for Training-free Methods

## E.1    Implementation of Baselines

In this subsection, we present the implementation details of training-free baselines, including LLM's direct inference, GraphSearch, and ours SGC. Method illustrations are shown in Figure 6.

**LLM's Direct Inference:** The prompt template for LLM's direct inference is given in Table 4. During experiments, we **uniformly** apply **1-shot in context learning** for LLM's direct inference of task invocation path. For open-sourced LLMs, the temperature parameter is set to 0.2.

**GraphSearch:** The prompt template for GraphSearch is given in Table 5. This algorithm conducts an iterative search on the task graph to identify an optimal task invocation path that can best satisfy a given request. In each iteration, the neighbors of the last selected task are considered as candidates. These candidates are evaluated by LLM for their suitability for the current step (*Task Assessment*). The search process follows a depth-first approach. After the task assessment in the final step, a set of potential invocation paths is generated. Subsequently, LLM is prompted to select the most appropriate path from these options (*Path Selection*). The GraphSearch algorithm is implemented in three distinct variants, each employing a unique task selection strategy:

- **GreedySearch** consistently selects the task node with the *highest* score at each step. Although fast and simple, this approach can lead to cascading errors, resulting in degraded performance.

- **AdaptiveSearch** selects tasks with scores *above a fixed threshold*, adjusting the breadth of the search space in an adaptive mode. During experiments, we empirically set the score threshold to 3.

- **BeamSearch** retains the *top-k* tasks based on the LLM's assessment scores within candidates. Beam search can expand the search space but slightly reduces the efficiency. We uniformly set the beam width to 2.

**Ours SGC:** Regarding the choices of LM backbones, for integrating GPT-3.5-turbo and GPT-4-turbo with SGC, the Roberta-355M [40] serves as the text encoder. For all other datasets and LLMs, the e5-335M [62] configuration is employed.

## E.2    Results of All LLMs

Table 8 supplements Table 1 with other LLMs. The proposed methods perform consistently better.

Table 8: **Comparison of Training-free Approaches: Overall Performance (Node-F1 and Link-F1 in %) and Token Consumption in $\times 10^3$.**

| LLM | Method | HuggingFace | | | TaskBench Multimedia | | | Daily Life | | | RestBench TMDB | | |
|---|---|---|---|---|---|---|---|---|---|---|---|---|---|
| | | n-F1 ↑ | l-F1 ↑ | # Tok ↓ | n-F1 ↑ | l-F1 ↑ | # Tok ↓ | n-F1 ↑ | l-F1 ↑ | # Tok ↓ | n-F1 ↑ | l-F1 ↑ | #Tok ↓ |
| **Baichuan2 13B** | Direct | 45.85 | 19.00 | 2.43 | 47.57 | 4.08 | 2.59 | 33.45 | 9.52 | 3.72 | 30.87 | 9.92 | 1.96 |
| | GreedySearch | 30.58 | 4.89 | 6.42 | 18.74 | 4.45 | 5.69 | 15.60 | 1.61 | 5.91 | 22.52 | 2.98 | 3.62 |
| | AdaptiveSearch | 39.30 | 10.41 | 10.81 | 33.24 | 9.22 | 11.17 | 34.39 | 12.73 | 16.71 | 30.33 | **10.00** | 8.71 |
| | BeamSearch | 41.06 | 9.59 | 24.69 | 32.24 | 9.09 | 21.60 | 36.18 | 13.18 | 23.83 | 30.97 | 7.61 | 9.08 |
| | SGC | **56.53** | **29.94** | **2.28** | **56.75** | **31.62** | **2.43** | **62.31** | **36.69** | **3.53** | **32.97** | 9.11 | **1.84** |
| **Vicuna 13B** | Direct | 50.46 | 21.27 | 2.50 | 53.57 | 23.19 | 2.64 | 73.70 | 45.80 | 3.82 | 44.66 | 14.01 | 2.02 |
| | GreedySearch | 52.94 | 25.73 | 6.23 | 46.99 | 23.11 | 5.55 | 42.98 | 13.33 | 7.18 | 45.22 | 13.69 | 3.42 |
| | AdaptiveSearch | 54.36 | 25.67 | 9.81 | 51.24 | 24.32 | 11.25 | 62.71 | 31.15 | 13.92 | 41.32 | 7.02 | 6.51 |
| | BeamSearch | 56.64 | 26.93 | 24.11 | 54.09 | 26.19 | 25.42 | 54.55 | 23.60 | 24.86 | 46.91 | 15.41 | 7.79 |
| | SGC | **59.62** | **31.98** | **2.31** | **61.78** | **37.60** | **2.43** | **83.33** | **63.77** | **3.82** | **48.79** | **15.99** | **1.89** |
| **CodeLlama 7B** | Direct | 58.06 | 29.39 | 2.44 | 59.44 | 30.83 | 2.57 | 84.12 | 62.89 | 3.82 | 65.67 | 41.99 | 1.94 |
| | GreedySearch | 58.71 | 31.56 | 5.84 | 62.83 | 38.12 | 5.35 | 82.51 | 63.83 | 7.08 | 65.51 | 42.60 | 3.12 |
| | AdaptiveSearch | 60.42 | 33.18 | 6.84 | 62.32 | 36.81 | 5.50 | 83.42 | 64.15 | 7.83 | 65.37 | 40.64 | 5.00 |
| | BeamSearch | 60.34 | 31.36 | 17.95 | 64.12 | 38.99 | 21.48 | 83.25 | 63.48 | 24.48 | 64.60 | 40.50 | 5.78 |
| | SGC | **63.98** | **39.27** | **2.30** | **67.04** | **45.04** | **2.43** | **87.73** | **70.49** | **3.59** | **66.15** | **42.62** | **1.88** |
| **Mistral 7B** | Direct | 60.60 | 30.23 | 2.49 | 69.83 | 39.85 | 2.64 | 84.26 | 53.63 | 3.77 | 62.23 | 22.02 | 1.96 |
| | GreedySearch | 65.91 | 38.13 | 6.52 | 58.92 | 34.72 | 6.26 | 75.18 | 49.47 | 8.27 | 60.64 | 23.18 | 4.38 |
| | AdaptiveSearch | 67.30 | 38.90 | 7.68 | 71.59 | 44.84 | 10.66 | 86.39 | 63.65 | 10.92 | 54.04 | 21.35 | 9.99 |
| | BeamSearch | 67.13 | 36.73 | 25.66 | 73.55 | 47.12 | 31.10 | 85.87 | 61.53 | 39.16 | 63.41 | **26.79** | 11.26 |
| | SGC | **67.43** | **42.08** | **2.32** | **74.07** | **49.90** | **2.43** | **87.13** | **66.49** | **3.54** | **64.72** | 25.67 | **1.89** |
| **CodeLlama 13B** | Direct | 57.55 | 28.88 | 2.45 | 68.57 | 41.79 | 2.59 | 91.20 | 76.07 | 3.88 | 68.91 | 43.74 | 2.02 |
| | GreedySearch | 61.67 | 34.02 | 5.95 | 67.98 | 42.04 | 4.95 | 91.50 | 76.56 | 5.54 | 66.67 | 42.16 | 3.81 |
| | AdaptiveSearch | 60.85 | 31.66 | 11.10 | 68.14 | 41.71 | 6.77 | 91.34 | 76.09 | 7.18 | 63.74 | 37.17 | 8.16 |
| | BeamSearch | 62.65 | 34.31 | 20.14 | 69.53 | 43.35 | 19.51 | 91.74 | 76.60 | 19.19 | 68.08 | 42.92 | 8.88 |
| | SGC | **65.51** | **39.44** | **2.31** | **73.32** | **53.28** | **2.43** | **92.96** | **79.57** | **3.64** | **71.40** | **47.55** | **1.90** |
| **GPT-3.5-turbo** | Direct | 73.85 | 45.73 | 2.14 | 82.85 | 62.07 | 2.26 | 96.09 | 83.65 | 3.36 | 81.70 | 57.52 | 1.67 |
| | GreedySearch | 67.75 | 43.88 | 5.29 | 81.11 | 63.02 | 4.92 | 93.77 | 81.26 | 7.36 | 76.19 | 50.11 | 3.06 |
| | AdaptiveSearch | 72.18 | 47.55 | 7.47 | 81.86 | 62.71 | 5.71 | 93.79 | 81.41 | 8.53 | 77.57 | 53.65 | 5.89 |
| | BeamSearch | 75.51 | 49.62 | 14.22 | 83.57 | **64.50** | 12.91 | 95.66 | 82.72 | 22.05 | 81.24 | 57.98 | 6.42 |
| | SGC | **76.37** | **50.04** | **2.02** | **83.65** | 63.65 | **2.09** | **96.38** | **86.19** | **3.16** | **82.63** | **59.15** | **1.61** |
| **GPT-4-turbo** | Direct | 77.60 | 52.18 | 2.19 | 88.29 | 69.38 | 2.28 | **97.36** | 84.58 | 3.37 | **82.56** | **56.67** | 1.75 |
| | GreedySearch | 74.75 | 50.44 | 5.78 | 86.81 | 69.80 | 5.52 | 97.36 | 84.78 | 7.37 | 75.34 | 49.95 | 3.73 |
| | AdaptiveSearch | 76.17 | 51.30 | 8.94 | 88.02 | 69.99 | 7.14 | 97.30 | **85.80** | 9.04 | 81.78 | 55.15 | 6.35 |
| | BeamSearch | 77.56 | **52.54** | 8.98 | 88.16 | **70.39** | 6.90 | 97.35 | 85.78 | 8.99 | 80.11 | 51.00 | 5.18 |
| | SGC | **77.79** | 52.20 | **2.03** | **88.54** | 69.83 | **2.10** | 97.35 | 85.76 | **3.16** | 82.27 | 56.37 | **1.62** |

Table 9: **Results of Supplementary Metric: Accuracy (%) for Training-free Methods on TaskBench**. Accuracy is 1 if predicted tasks match the ground-truth task set, and 0 otherwise.

| LLM | Method | HuggingFace | Multimedia | DailyLife | LLM | Method | HuggingFace | Multimedia | DailyLife |
|---|---|---|---|---|---|---|---|---|---|
| **Vicuna 13B** | Direct | 8.72 | 11.20 | 24.43 | **CodeLlama 7B** | Direct | 15.00 | 15.19 | 47.69 |
| | GreedySearch | 10.95 | 9.34 | 3.76 | | GreedySearch | 16.20 | 20.04 | 45.07 |
| | AdaptiveSearch | 10.55 | 10.37 | 13.15 | | AdaptiveSearch | 18.79 | 19.41 | 46.48 |
| | BeamSearch | 12.58 | 12.03 | 11.06 | | BeamSearch | 17.00 | 21.10 | 45.67 |
| | SGC | **16.02** | **20.12** | **42.17** | | SGC | **21.20** | **29.32** | **55.33** |
| **Mistral 7B** | Direct | 16.36 | 25.05 | 44.52 | **CodeLlama 13B** | Direct | 14.29 | 24.10 | 66.40 |
| | GreedySearch | 20.45 | 16.02 | 29.22 | | GreedySearch | 19.11 | 24.90 | 67.00 |
| | AdaptiveSearch | 21.88 | 26.90 | 49.32 | | AdaptiveSearch | 17.30 | 24.10 | 66.80 |
| | BeamSearch | 20.45 | 29.36 | 45.89 | | BeamSearch | 19.92 | 25.70 | 67.20 |
| | SGC | **25.15** | **33.68** | **52.28** | | SGC | **22.54** | **36.75** | **70.80** |
| **GPT-3.5-turbo** | Direct | 28.95 | 47.96 | 81.30 | **GPT-4-turbo** | Direct | 33.68 | 60.56 | **86.77** |
| | GreedySearch | 26.90 | **52.47** | 73.17 | | GreedySearch | 33.68 | 61.37 | 86.77 |
| | AdaptiveSearch | 29.36 | 51.61 | 74.59 | | AdaptiveSearch | 33.47 | 61.17 | 86.77 |
| | BeamSearch | 32.03 | 52.47 | 80.87 | | BeamSearch | 33.26 | **61.57** | 86.57 |
| | SGC | **32.44** | 51.61 | **83.13** | | SGC | **34.09** | 60.97 | 86.77 |

## E.3 Accuracy Results of Training-free Methods

Due to space limitations, we present only the Node-F1 and Link-F1 scores for training-free methods in the main text. Here, we provide the Accuracy results in Table 9. These results show that integrating SGC significantly enhances accuracy across different LLMs on all datasets, making previously unsolvable planning scenarios manageable and successful.

## E.4 Computational Cost Analysis

In this subsection, we present a comprehensive efficiency study on inference time of training-free methods and results are shown in Table 10.

Table 10: **Computational Cost Analysis of Training-free Methods.** Due to space constraints in the table, some LLMs are abbreviated such as "GPT-3.5" for "GPT-3.5-turbo".

| Method | Inference Times Comparison on HuggingFace (Seconds) | | | | | | |
|---|---|---|---|---|---|---|---|
| | Baichuan | Vicuna | CodeLlama-7B | Mistral | CodeLlama-13B | GPT-3.5 | GPT-4 |
| Direct | 6.0 | 3.6 | 10.9 | 4.5 | 9.7 | 2.7 | 26.1 |
| GreedySearch | 30.7 | 45.7 | 23.1 | 109.8 | 29.1 | 7.4 | 55.6 |
| AdaptiveSearch | 50.2 | 79.4 | 27.2 | 28.2 | 52.4 | 9.4 | 87.0 |
| BeamSearch | 102.0 | 55.0 | 60.8 | 85.5 | 92.3 | 14.9 | 270.2 |
| **SGC** | 6.1 | 3.7 | **10.7** | 4.6 | **9.5** | 3.0 | **24.4** |
| | Inference Times Comparison on Multimedia (Seconds) | | | | | | |
| Direct | 9.7 | 3.3 | 13.2 | 4.5 | 14.6 | 2.9 | 25.1 |
| GreedySearch | 52.3 | 54.3 | 37.0 | 109.7 | 9.9 | 8.8 | 52.2 |
| AdaptiveSearch | 98.1 | 142.5 | 25.7 | 41.6 | 25.6 | 9.6 | 84.2 |
| BeamSearch | 122.3 | 69.8 | 103.0 | 92.0 | 84.4 | 15.0 | 70.9 |
| **SGC** | **9.5** | 3.4 | **12.9** | **4.5** | **14.1** | 3.1 | **23.5** |
| | Inference Times Comparison on Daily Life (Seconds) | | | | | | |
| Direct | 10.0 | 6.5 | 19.4 | 5.6 | 18.6 | 3.4 | 31.0 |
| GreedySearch | 62.7 | 49.8 | 29.3 | 198.3 | 37.6 | 13.2 | 124.3 |
| AdaptiveSearch | 133.3 | 97.6 | 30.5 | 69.1 | 45.5 | 16.5 | 209.0 |
| BeamSearch | 196.9 | 54.1 | 106.9 | 195.9 | 64.9 | 89.4 | 161.7 |
| **SGC** | **9.9** | **6.5** | **18.6** | 5.7 | **17.9** | 3.6 | **29.5** |

Open-sourced LLMs were deployed as local API services using the FastChat framework[4] on a single A100-80G GPU. This configuration enables faster and parallel inference. Under this setup, LLM's direct inference **requires 3-15 seconds** per request. GPT-3.5-turbo and GPT-4-turbo are accessed via API, with the latter generally requiring more time. GraphSearch requires **several minutes** to complete a request due to its exhaustive search on the task graph, impacting the efficiency. In contrast, SGC achieves comparable efficiency to LLM's direct inference, as it requires only a single LLM query and both LM and SGC's forward propagation processes are extremely efficient (typically completing within seconds). Note that some discrepancies in reported times, such as Mistral-7B's GreedySearch taking longer than other modes, may be attributed to variations in the deployment across different A100 services.

---

[4]https://github.com/lm-sys/FastChat

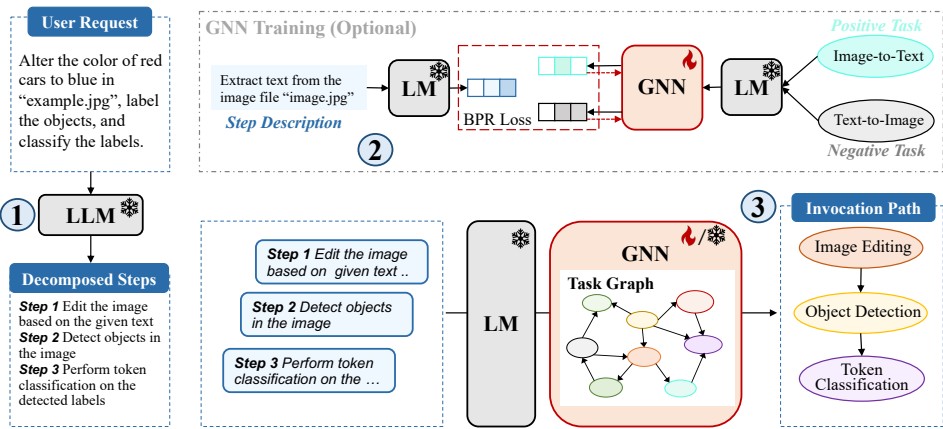

Figure 7: **Illustration of our GNN-enhanced Task Planning.** First, **LLMs interpret user request into several manageable steps**. Then, we **leverage GNNs for task retrieval**, sequentially matching each step description to a suitable task, finally generating the invocation path.

# F Supplementary Materials for Training-based Methods

## F.1 Implementation of Training-based GNNs

**LM and GNN Configuration:** For training-based GNNs, we uniformly use the e5-335M model [62] as the LM backbone. For the graph encoder, our setup includes a single layer with a hidden dimension of $1024$. During the model training, we set the batch size to $512$ and run for $20$ epochs with a learning rate of $1e-3$. We use the Adam optimizer [25] and implement an early stopping mechanism with a patience of $5$ epochs to prevent over-fitting. All experiments are conducted on a single NVIDIA A100-80G GPU.

**Training Data Preparation:** From each dataset in TaskBench, we randomly select $3,000$ samples to create the trainset. The original data includes specific task steps and corresponding task invocation paths. Therefore, we first employ a topological sort to align each task step accurately with corresponding task, forming "<step, ground-truth task>" pairs. Then, for each pair, we randomly sample two negative tasks to constitute the "**<step, positive task, negative task>**" triplets for model training. These negative samples are selected based on how textually similar they are to the positive one, creating a robust differentiation challenge for the model.

**Choices of Different Configurations:** Our training-based model offers two configuration options: training only the GNN while keeping the LM frozen, or co-training both the LM and GNN. Illustrations of these configurations are provided in Figure 8. The first configuration is designed to explore the GNN's capability in task retrieval. The latter leverages the LM's dataset-specific semantic embeddings to enhance performance. For the co-training setup, we use a learning rate of $2e-5$ and a training duration of $10$ epochs.

## F.2 Implementation of TAPE and GraphToken

We adapt TAPE [16] and GraphToken [38] as training-required baselines for task planning. Here, we detail the adaptation processes for each method.

**TAPE:** To adapt TAPE for task planning, we reformulate the planning task as a node classification problem, aiming to classify a user request into the appropriate task labels within the task graph. Firstly, LLMs interpret user requests by generating high-quality explanatory text, i.e., chain-of-thought reasoning, to understand each request. Then, a fine-tuned LM encodes these chain-of-thought texts into latent embeddings. The fine-tuning process is based on pairs of textual descriptions and their corresponding ground-truth tasks. Finally, GNNs select the suitable tasks by leveraging both the generated text embeddings and the task embeddings. For a fair performance comparison, we fine-tune the LM as e5-335M and configure the GNN as a 2-layer GraphSAGE with a hidden dimension of

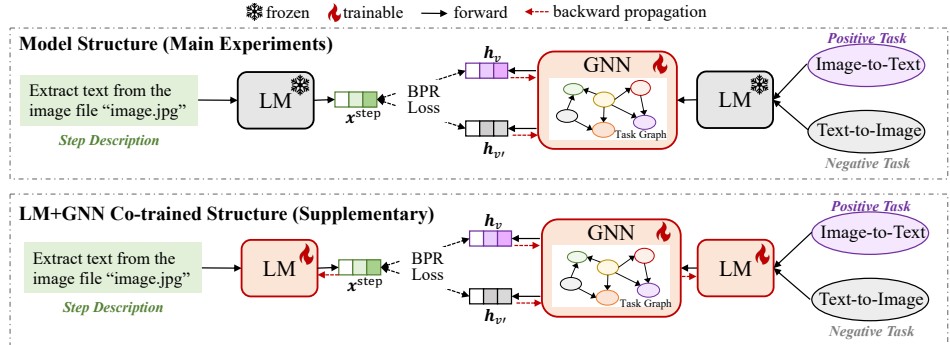

Figure 8: **LM+GNN Configuration**. We offer two configurations: only training the GNN while keeping the LM frozen (Table 11), or co-training both the LM and GNN (Table 12).

1024. The training data for both the LM and GNNs are consistent, utilizing the same split of the training dataset.

**GraphToken:** For adapting GraphToken, we first encode the task graph $G(V, E, X)$ by computing each node's representation $x_v$ by feeding its descriptive text into the pre-trained LM, e5-335M. Then, a GNN transforms the initial node embedding matrix into $H$. The GNN is configured as a 2-layer GraphSAGE with a hidden dimension of 1024. A mean pooling operation is further applied to the node embeddings to obtain the graph's overall representation as $h_G = \text{Mean-Pooling}(H)$. Finally, both $h_G$ and the text embeddings of input instruction and user request are concatenated and fed into the LLM to generate the output planning result. During this process, the backbone LLMs are frozen, and only the GNN's parameters are tuned. For each dataset, we train over 4 epochs using the same 3,000 training samples consisting of <user request, ground-truth planning> pairs. The batch size is set to 16, with a maximum input length of 512 and a maximum output length of 300 for HuggingFace and Multimedia datasets, and 600 for DailyLife dataset. The learning rate is configured to $1e - 5$.

## F.3 Implementation of Fine-tuning LLMs

To explore the effectiveness of our proposed framework on fine-tuned LLMs, we employ Supervised Fine-Tuning (SFT) on the CodeLlama-7B and Vicuna-13B models. We use **the same set of** 3,000 **samples** from GNN training as the LLMs' fine-tuning data. In fine-tuning the LLM with LoRA [19], we set the `lora_r` parameter (dimension for LoRA update matrices) to 8 and the `lora_alpha` (scaling factor) to 16. The dropout ratio is set to 0.1, the batch size to 2, and we conduct training over 2 epochs with a learning rate of $1e - 5$. For HuggingFace dataset, the maximum input length is set to 800, while the maximum output length is 400. For Multimedia and Daily Life datasets, which contain a larger number of tasks and require longer textual inputs, we set the maximum input and output length to 1000 and 500, respectively. We utilize 2 NVIDIA A100-80G GPUs for fine-tuning the LLMs.

## F.4 Full Results of Training-based GNNs

In Table 11, we present comprehensive results for all training-based GNNs, including GCN [26], GAT [57], GraphSAGE [15], GIN [67], and Graph Transformer [47]. To highlight the improvements brought by GNNs, we also include results from the strongest baseline, BeamSearch.

From the results, it is obvious that **all GNN encoders significantly enhance the task planning abilities**. For instance, when applied to Vicuna-13B on HuggingFace dataset, the introduction of GCN results in a performance improvement of 17.83%, GAT contributes to a 17.48% increase, GraphSAGE leads to a 22.59% boost, and GraphTransformer improves predictions by 18.0%. This conclusion can be generalized across various LLMs and datasets, demonstrating the robust capabilities of GNNs.

Table 11: **Performance of training-based GNNs**. We also presents the results of BeamSeaerch, the strongest variant from GraphSearch method to provide a comprehensive comparison. All GNNs consistently enhance task planning performance across diverse LLMs, showing the effectiveness.

| LLM | Method | HuggingFace Tools | | Multimedia Tools | | Daily Life APIs | |
| | | $n\text{-}F1\uparrow$ | $l\text{-}F1\uparrow$ | $n\text{-}F1\uparrow$ | $l\text{-}F1\uparrow$ | $n\text{-}F1\uparrow$ | $l\text{-}F1\uparrow$ |
|---|---|---|---|---|---|---|---|
| **Baichuan-13B** | Direct | 45.85 | 19.00 | 47.57 | 4.08 | 33.45 | 9.52 |
| | BeamSearch | 41.06 | 9.59 | 32.24 | 9.09 | 36.18 | 13.18 |
| | GCN | 57.67 | 31.47 | 55.51 | 30.16 | 62.11 | 37.05 |
| | GAT | 57.74 | 31.87 | 54.95 | 29.24 | 62.11 | 37.05 |
| | GraphSAGE | **59.32** | **34.36** | **56.15** | **31.60** | **65.18** | **40.49** |
| | GIN | 57.38 | 31.17 | 55.08 | 30.04 | 62.11 | 37.05 |
| | GraphTransformer | 59.15 | 34.36 | 56.06 | 31.12 | 64.52 | 40.14 |
| **Vicuna-13B** | Direct | 50.46 | 21.27 | 53.57 | 23.19 | 73.70 | 45.80 |
| | BeamSearch | 56.64 | 26.93 | 54.09 | 26.19 | 54.55 | 23.60 |
| | GCN | 59.46 | 33.14 | 62.48 | 38.89 | 83.05 | 62.95 |
| | GAT | 59.28 | 33.39 | 62.96 | 39.24 | 83.05 | 62.95 |
| | GraphSAGE | **61.86** | **35.68** | **63.71** | **39.88** | **86.07** | **67.63** |
| | GIN | 59.14 | 32.33 | 62.61 | 38.82 | 83.05 | 62.95 |
| | GraphTransformer | 59.57 | 33.47 | 63.32 | 39.38 | 85.41 | 66.28 |
| **CodeLlama-7B** | Direct | 58.06 | 29.39 | 59.44 | 30.83 | 84.12 | 62.89 |
| | BeamSearch | 60.34 | 31.36 | 64.12 | 38.99 | 83.25 | 63.48 |
| | GCN | 65.07 | 40.50 | 67.46 | 45.84 | 87.23 | 69.27 |
| | GAT | 65.20 | 40.93 | 67.41 | 46.46 | 87.23 | 69.27 |
| | GraphSAGE | **66.67** | **43.03** | 67.97 | 46.31 | **88.53** | **72.02** |
| | GIN | 65.52 | 40.98 | 66.89 | 45.41 | 87.23 | 69.27 |
| | GraphTransformer | 65.83 | 42.58 | **68.90** | **47.20** | 88.36 | 71.72 |
| **Mistral-7B** | Direct | 60.60 | 30.23 | 69.83 | 39.85 | 84.26 | 53.63 |
| | BeamSearch | 67.13 | 36.73 | 73.55 | 47.12 | 85.87 | 61.53 |
| | GCN | 66.54 | 40.74 | 73.34 | 50.76 | 86.39 | 65.49 |
| | GAT | 66.77 | 40.74 | 73.36 | 50.20 | 86.39 | 65.49 |
| | GraphSAGE | 68.12 | **43.09** | **75.51** | **52.94** | 87.51 | 66.57 |
| | GIN | 66.69 | 40.79 | 72.89 | 50.44 | 86.39 | 65.49 |
| | GraphTransformer | **68.26** | 43.08 | 73.80 | 51.45 | **88.25** | **67.84** |
| **CodeLlama-13B** | Direct | 57.55 | 28.88 | 68.57 | 41.79 | 91.20 | 76.07 |
| | BeamSearch | 62.65 | 34.31 | 69.53 | 43.35 | 91.74 | 76.60 |
| | GCN | 66.22 | 41.05 | 72.99 | 52.18 | 91.83 | 77.88 |
| | GAT | 66.29 | 41.28 | 74.08 | 53.56 | 91.83 | 77.88 |
| | GraphSAGE | **67.30** | **42.41** | **74.93** | **54.52** | **93.84** | 80.38 |
| | GIN | 66.40 | 40.89 | 73.62 | 53.15 | 91.83 | 77.88 |
| | GraphTransformer | 66.70 | 42.07 | 74.72 | 54.10 | 93.81 | **80.44** |
| **GPT-3.5-turbo** | Direct | 73.85 | 45.73 | 82.85 | 62.07 | 96.09 | 83.65 |
| | BeamSearch | 75.51 | 49.62 | 83.57 | 64.50 | 95.66 | 82.72 |
| | GCN | 76.93 | 51.43 | 84.92 | 65.05 | 96.38 | 86.15 |
| | GAT | 75.63 | 49.36 | 84.77 | 65.48 | 96.38 | 86.15 |
| | GraphSAGE | **77.90** | **52.68** | **85.29** | **65.80** | **96.43** | **86.26** |
| | GIN | 76.86 | 51.00 | 84.14 | 64.30 | 96.38 | 86.15 |
| | GraphTransformer | 77.61 | 52.30 | 84.21 | 64.32 | 96.38 | 86.19 |
| **GPT-4-turbo** | Direct | 77.60 | 52.18 | 88.29 | 69.38 | 97.36 | 84.58 |
| | BeamSearch | 77.56 | **52.54** | 88.16 | **70.39** | 97.35 | **85.78** |
| | GCN | 77.01 | 50.49 | 88.56 | 69.60 | 97.10 | 85.22 |
| | GAT | 76.41 | 49.66 | 88.43 | 69.52 | 97.10 | 85.22 |
| | GraphSAGE | **78.76** | 52.53 | **88.63** | 69.65 | 97.34 | 85.67 |
| | GIN | 77.74 | 51.02 | 88.05 | 69.13 | **97.36** | 84.58 |
| | GraphTransformer | 78.47 | 52.17 | 88.07 | 68.71 | 97.32 | 85.57 |

Table 12: **Performance of Training-based GNNs under the LM+GNN Co-trained Mode**. Simultaneous training of LM and GNN yields significant performance improvements.

| LLM | Method | HuggingFace Tools | | Multimedia Tools | | Daily Life APIs | |
|---|---|---|---|---|---|---|---|
| | | *n-F1* ↑ | *l-F1* ↑ | *n-F1* ↑ | *l-F1* ↑ | *n-F1* ↑ | *l-F1* ↑ |
| **Baichuan-13B** | Direct | 45.85 | 19.00 | 47.57 | 4.08 | 33.45 | 9.52 |
| | SGC | **60.97** | 36.12 | 56.02 | 31.36 | 64.84 | 40.00 |
| | GCN | 60.68 | 36.31 | 57.82 | 32.87 | 64.73 | 38.92 |
| | GAT | 60.39 | 35.37 | 57.24 | 32.19 | 64.46 | 40.14 |
| | GraphSAGE | 59.76 | 35.59 | **57.97** | **33.29** | 63.21 | 38.10 |
| | GIN | 60.31 | 35.82 | 56.62 | 31.53 | 63.55 | 38.19 |
| | GraphTransformer | 60.76 | **36.82** | 56.82 | 31.40 | **64.88** | **40.23** |
| **Vicuna-13B** | Direct | 50.46 | 21.27 | 53.57 | 23.19 | 73.70 | 45.80 |
| | SGC | **64.40** | **38.97** | 65.12 | 41.63 | 84.74 | 65.90 |
| | GCN | 62.06 | 35.49 | 65.02 | 40.63 | 85.22 | 66.93 |
| | GAT | 63.06 | 36.97 | 64.58 | 40.30 | **85.63** | **67.11** |
| | GraphSAGE | 62.82 | 37.04 | **65.89** | **42.18** | 84.23 | 65.44 |
| | GIN | 62.09 | 35.33 | 64.44 | 40.67 | 85.31 | 66.83 |
| | GraphTransformer | 62.11 | 36.01 | 64.57 | 40.17 | 85.42 | 66.55 |
| **CodeLlama-7B** | Direct | 58.06 | 29.39 | 59.44 | 30.83 | 84.12 | 62.89 |
| | SGC | **67.47** | **43.58** | 69.61 | 48.24 | 87.98 | 70.63 |
| | GCN | 67.03 | 43.24 | 69.33 | 47.60 | 87.88 | 70.40 |
| | GAT | 67.12 | 42.96 | 68.62 | 46.17 | **88.59** | **71.64** |
| | GraphSAGE | 67.19 | 42.94 | **70.00** | **48.28** | 87.81 | 70.20 |
| | GIN | 66.62 | 42.34 | 69.00 | 47.72 | 88.45 | 71.53 |
| | GraphTransformer | 67.12 | 43.08 | 69.27 | 47.96 | 88.43 | 71.59 |
| **Mistral-7B** | Direct | 60.60 | 30.23 | 69.83 | 39.85 | 84.26 | 53.63 |
| | SGC | **69.04** | **44.22** | 76.09 | 54.91 | 87.58 | 66.70 |
| | GCN | 67.72 | 43.02 | 76.79 | 54.90 | **87.87** | 67.13 |
| | GAT | 67.54 | 43.56 | 76.26 | 53.94 | 87.86 | **67.30** |
| | GraphSAGE | 67.61 | 43.14 | 76.96 | **55.46** | 87.61 | 66.75 |
| | GIN | 68.95 | 43.97 | 76.47 | 54.95 | 87.75 | 67.07 |
| | GraphTransformer | 67.94 | 43.52 | **77.06** | 55.39 | 87.76 | 67.00 |
| **CodeLlama-13B** | Direct | 57.55 | 28.88 | 68.57 | 41.79 | 91.20 | 76.07 |
| | SGC | **70.14** | 45.20 | 75.65 | **55.45** | 93.45 | 79.89 |
| | GCN | 69.39 | 45.18 | 76.03 | 55.22 | 93.38 | 79.74 |
| | GAT | 69.68 | **45.57** | 75.24 | 54.99 | 94.06 | 80.96 |
| | GraphSAGE | 68.92 | 44.85 | **76.28** | 55.41 | 93.30 | 79.51 |
| | GIN | 69.01 | 44.76 | 74.72 | 53.91 | **94.24** | **81.23** |
| | GraphTransformer | 69.52 | 45.68 | 75.46 | 55.14 | 93.98 | 81.06 |
| **GPT-3.5-turbo** | Direct | 73.85 | 45.73 | 82.85 | 62.07 | 96.09 | 83.65 |
| | SGC | 77.87 | 52.86 | 85.95 | 66.95 | **96.39** | 86.16 |
| | GCN | 77.72 | 52.58 | 85.84 | 66.92 | 96.33 | 86.06 |
| | GAT | 77.49 | 52.30 | 85.81 | 66.97 | 96.38 | 86.15 |
| | GraphSAGE | **77.87** | **53.04** | 85.51 | 66.56 | 96.34 | 86.09 |
| | GIN | 77.73 | 52.36 | 85.63 | 66.69 | 96.38 | **86.19** |
| | GraphTransformer | 77.78 | 52.79 | **86.09** | **67.26** | 96.33 | 86.06 |
| **GPT-4-turbo** | Direct | 77.60 | 52.18 | 88.29 | 69.38 | 97.36 | 84.58 |
| | SGC | 78.44 | 52.84 | **89.09** | **70.52** | 97.38 | **85.85** |
| | GCN | 78.33 | 52.75 | 89.00 | 70.24 | 97.34 | 85.67 |
| | GAT | 78.37 | 52.43 | 88.99 | 70.48 | 97.32 | 85.56 |
| | GraphSAGE | **78.49** | 52.62 | 88.86 | 70.25 | **97.42** | 85.80 |
| | GIN | 78.45 | **53.07** | 88.74 | 69.84 | 97.42 | 85.80 |
| | GraphTransformer | 78.30 | 52.27 | 88.90 | 70.24 | 97.42 | 85.80 |

## F.5 Performance of LM+GNN Co-trained Mode

During our main experiments, the LM backbone remains frozen while only the GNN is trained to automatically learn the alignment between implicit step descriptions and suitable tasks, facilitating task retrieval. In this subsection, we conduct a supplementary study where the parameters of a pre-trained LM are also tuned along with GNN during model training. The model configuration is illustrated in Figure 8, and the results are presented in Table 12.

The results demonstrate that, compared to the GNN-only tunable mode, co-training LM+GNN can lead to further performance improvements. This enhancement occurs because the language model acquires task-specific semantics, which makes the representations more discriminative and boosts the GNN's effectiveness in task retrieval. Additionally, it is noted that under the co-training setup, the differences between various GNN encoders are relatively minor, with performance variations across GNNs for a specific LLM on any dataset remaining within $2\%$.

## F.6 Computational Cost Analysis

In this subsection, we provide the computational costs of training-based methods: training time for GNN or LM+GNN co-trained, and resources needed for fine-tuning LLMs. Results are shown in Table 13.

Table 13: **Computational Cost Analysis of Training-based Methods.** We present total training times for both GNN and LM+GNN co-trained modes, and resources needed for fine-tuning LLMs.

| Training GNNs | | | | | |
|---|---|---|---|---|---|
| Mode | Configuration | # Parameters | Time (Seconds) | | |
| | | | HuggingFace | Multimedia | Daily Life |
| GNN-only | GCN | 1,049,600 | 136.5 | 136.1 | 237.7 |
| | GAT | 1,051,648 | 136.9 | 151.8 | 237.8 |
| | GraphSAGE | 2,098,176 | 134.2 | 149.8 | 233.8 |
| | GIN | 2,099,200 | 134.2 | 134.9 | 233.6 |
| | GraphTransformer | 4,198,400 | 135.0 | 150.2 | 233.7 |
| LM+GNN Co-trained | LM+SGC | 335,141,889 | 743.1 | 323.3 | 384.7 |
| | LM+GCN | 336,191,488 | 482.8 | 567.8 | 384.2 |
| | LM+GAT | 336,193,536 | 741.3 | 812.3 | 384.4 |
| | LM+GraphSAGE | 337,240,064 | 741.2 | 406.8 | 381.7 |
| | LM+GIN | 337,241,088 | 735.6 | 361.8 | 382.9 |
| | LM+GraphTransformer | 339,340,288 | 741.0 | 362.7 | 405.8 |

| Fine-tuning LLMs | | | | |
|---|---|---|---|---|
| LLM | # Param (# Trainable Param) | Device & Time (Hours) | | |
| | | HuggingFace | Multimedia | Daily Life |
| CodeLlama-7B | 6,742,740,992 (4,194,304) | 2×A100 7.0 | 1×A100 13.8 | 2×A100 9.5 |
| Vicuna-13B | 13,022,417,920 (6,553,600) | 1×A100 17.8 | 2×A100 10.3 | 2×A100 19.0 |

**Efficiency of Training GNNs:** During experiments, each dataset shares the same GNN configuration: 1 single layer with a hidden dimension of 1024. Therefore, for each GNN, its number of parameters remains consistent across datasets. The parameter scales for GNN variants range from 1M to 4M, and the total training time for each dataset requires only **2-4 minutes**, comparable to the time taken by GraphSearch to fulfill a single request. For LM+GNN co-trained mode, where e5-335M [62] serves as the LM backbone, training times increase to approximately **6-12 minutes**. In summary, both modes demonstrate high efficiency, with total training times spanning minutes, showcasing their ability to rapidly adapt to new task planning scenarios.

**Efficiency of Fine-tuning LLMs:** Fine-tuning a LLM with $3,000$ training samples over 2 epochs requires huge time, typically **10-20 hours** on one or two A100-80G GPU devices.

# G   Experiments on Task Parameter Prediction

To assess the quality of task planning, we primarily focus on the predicted tasks and their dependencies, leaving the parameters for invoking these tasks undiscussed. As LLMs' direct inference for solely predicting tasks has proven unsatisfactory (Figure 2), relying on LLMs alone to directly predict these parameters is unreliable. In this section, we will first demonstrate that it is quite straightforward for LLMs to fill in the parameters given a planned task sequence, which is a simple extension of our framework. Furthermore, we will empirically show that with more accurately planned tasks, i.e., those retrieved by GNNs, LLMs can intelligently fill in the parameters.

## G.1   Prompting LLMs to Fill in Parameters

**Process:** Recall that our framework enables more accurate invoked task sequences, e.g., $\{v_1, \ldots, v_n\}$, for a specific user request. To finalize the invocation sequence, just adding an additional LLM query can complete the invocation parameters for each task. Specifically, by providing the original user request, planned tasks, and detailed descriptions including each task's input and output requirements, LLMs can be intelligently prompted to fill in the invocation parameters for each planned task, resulting in a well-structured invocation sequence ready for execution. Prompts are provided in Table 6.

**Example:** Taking the request shown in Figure 1 as an example, which is: "Please generate an image where a girl is reading a book, and her pose is the same as the boy in 'example.jpg'. Then, please describe the new image with your voice." Suppose GNN's planned tasks include { `Pose Detection`, `Pose-to-Image`, `Image-to-Text`, `Text-to-Speech` }. LLMs can intelligently fill in the invocation arguments as follows: [ {"task": `Pose Detection`, "arguments": ['example.jpg'] }, {"task": `Pose-to-Image`, "arguments": [output pose from `Pose Detection`] }, {"task": `Image-to-Text`, "arguments": [output image from `Pose-to-Image`] }, {"task": `Text-to-Speech`, "arguments": [output text from `Image-to-Text`]}].

## G.2   Empirical Results of LLMs Predicted Parameters

**Experimental Setup:** We conduct this supplementary experiment on the HuggingFace dataset from TaskBench [45]. Specifically, we compare the directly predicted invocation parameters from LLMs (denoted as **Direct**) with the parameters filled automatically by LLMs using our GNN-retrieved tasks (querying prompt as shown in Table 6). For GNNs, we consider both training-free SGC and training-required GraphSAGE. Notably, with SGC, since the query for completing parameters is also training-free, the entire pipeline can be deployed without any extensive model training or labeled data. We adopt evaluation metrics from TaskBench, including **Parameter-Type F1-Score** (*Param t-F1*) and **Parameter-Value F1-Score** (*Param v-F1*). These metrics measure the accuracy of the predicted parameter types and concrete values, respectively. For example, when filling in the invocation for `Pose Detection`, the ground-truth parameter type is `Image`, and the ground-truth value is 'example.jpg'.

Table 14: **Performance (in %) of Task Parameters Prediction on the HuggingFace dataset**

| LLM | Method | t-F1 ↑ | v-F1 ↑ | LLM | Method | t-F1 ↑ | v-F1 ↑ |
|---|---|---|---|---|---|---|---|
| **Mistral-7B** | Direct | 38.77 | 18.56 | **CodeLlama-13B** | Direct | 44.62 | 33.24 |
| | SGC | 58.13$_{+19.36}$ | 39.64$_{+21.08}$ | | SGC | 57.74$_{+13.21}$ | 43.21$_{+9.97}$ |
| | GraphSAGE | **59.07**$_{+20.30}$ | **41.40**$_{+22.84}$ | | GraphSAGE | **59.49**$_{+14.87}$ | **45.54**$_{+12.30}$ |
| **GPT-3.5-turbo** | Direct | 62.42 | 48.27 | **GPT-4-turbo** | Direct | 70.73 | 55.54 |
| | SGC | 68.13$_{+5.71}$ | 54.34$_{+6.07}$ | | SGC | 72.91$_{+2.18}$ | 58.02$_{+2.48}$ |
| | GraphSAGE | **71.19**$_{+8.77}$ | **56.83**$_{+8.56}$ | | GraphSAGE | **73.09**$_{+2.36}$ | **58.20**$_{+2.66}$ |

**Observation:** From the results shown in Table 14, we can conclude that: (1) LLMs' direct inference of invocation arguments is unsatisfactory. Even for the strongest LLM, GPT-4-turbo, the Parameter-Value F1 score is only 55%, which is far from expectations. (2) With accurate planning provided by GNNs, LLMs can leverage their inherent reasoning abilities to analyze the context and correctly fill in the parameters. Across four LLMs, improvements of 3% to 23% can be observed, highlighting the advantages of GNN-enhanced planning. (3) Our method is well-suited for training-free scenarios, as the tasks retrieved by the training-free SGC already enable LLMs to infer the parameters effectively, with only a minor performance gap compared to the training-required GraphSAGE.

# H Case Studies

**Effectiveness of GNNs:** We show two cases in Figure 9 where the results of LLM's direct inference, BeamSearch, and GraphSAGE are compared. Due to space constraints and issues such as LLM's output content decoding errors or invalid paths, we present only the first four valid paths searched by BeamSearch on the task graph. From the cases, we can conclude that BeamSearch relies on LLM's inherent reasoning abilities. Although LLM can explore the ground-truth invocation path on the task graph, their final solutions are usually not the optimal as either containing the hallucination or wrongly invoked tasks due to limited instruction following and reasoning abilities. On the contrary, GNN can effectively align decomposed steps with suitable tasks, accurately achieving the ground-truth result and enhancing task planning.

**Failure Cases of GNNs:** We also present the failure cases where GNN performance deteriorates compared to direct inference to provide a comprehensive discussion of our method. Our conclusion is that the method is sensitive to the quality of decomposed task steps, as an ambiguous step may mislead GNN to wrongly select the task, and such errors can cascade due to the sequential selection of tasks on the task graph. As the case shown in Figure 10, step 2 is ambiguously described as "Segment the image and identify the tabular data", which actually incorporates two distinct steps. This ambiguity causes GNN to choose the unsuitable `Tabular Classification` instead of the correct `Image Segmentation`. Since tasks are selected sequentially on the task graph, where the next task is a neighbor of the current selection, such an error can prevent the exploration of the next appropriate task, as it may not be a neighbor of the current, incorrect selection. We also present the BeamSearch explored paths and its final solution, where it hits the ground-truth result.

**Diagnosing GraphSearch with GPT-4-turbo:** In our experiments, GraphSearch brings marginal or even decreased performance for GPT-4-turbo across most experimental datasets, contradicting current research [33, 32] which suggests that exhaustive search strategies can enhance the performance of more powerful LLMs. To provide a detailed analysis, we show three types of cases in Figure 11: Successful cases, Failure cases, as well as Maintaining cases:

- **Successful Cases:** As shown in the figure, despite inaccuracies in task decomposition (GPT-4 predicts an extra step compared to the three-step ground truth), its inherent reasoning abilities and the knowledge explored along the task graph can fix these mistakes, hitting the ground-truth.

- **Failure Cases:** In the failure case, although GPT-4 identifies the ground-truth solution during searching process, the final decision includes an incorrect task. This occurs because the final solution selection **demands complex reasoning abilities of LLM**, as the context contains long textual information from different aspects: **the full task graph, user request, all searched paths, and related instructions**. The demanding context and reasoning challenge exceed GPT-4's capabilities, leading to errors.

- **Maintaining Cases:** These occur when GraphSearch merely replicates the result of LLM's direct inference, indicating no added benefit from exploring the task graph. In these instances, despite navigating the graph, the LLM fails to self-correct inaccuracies due to inherent reasoning limitations.

We emphasize that the results of GraphSearch with GPT-4-turbo, tend to "maintaining" cases. Under such conditions, even a few failures can degrade overall performance, explaining why GraphSearch does not consistently enhance performance for GPT-4-turbo.

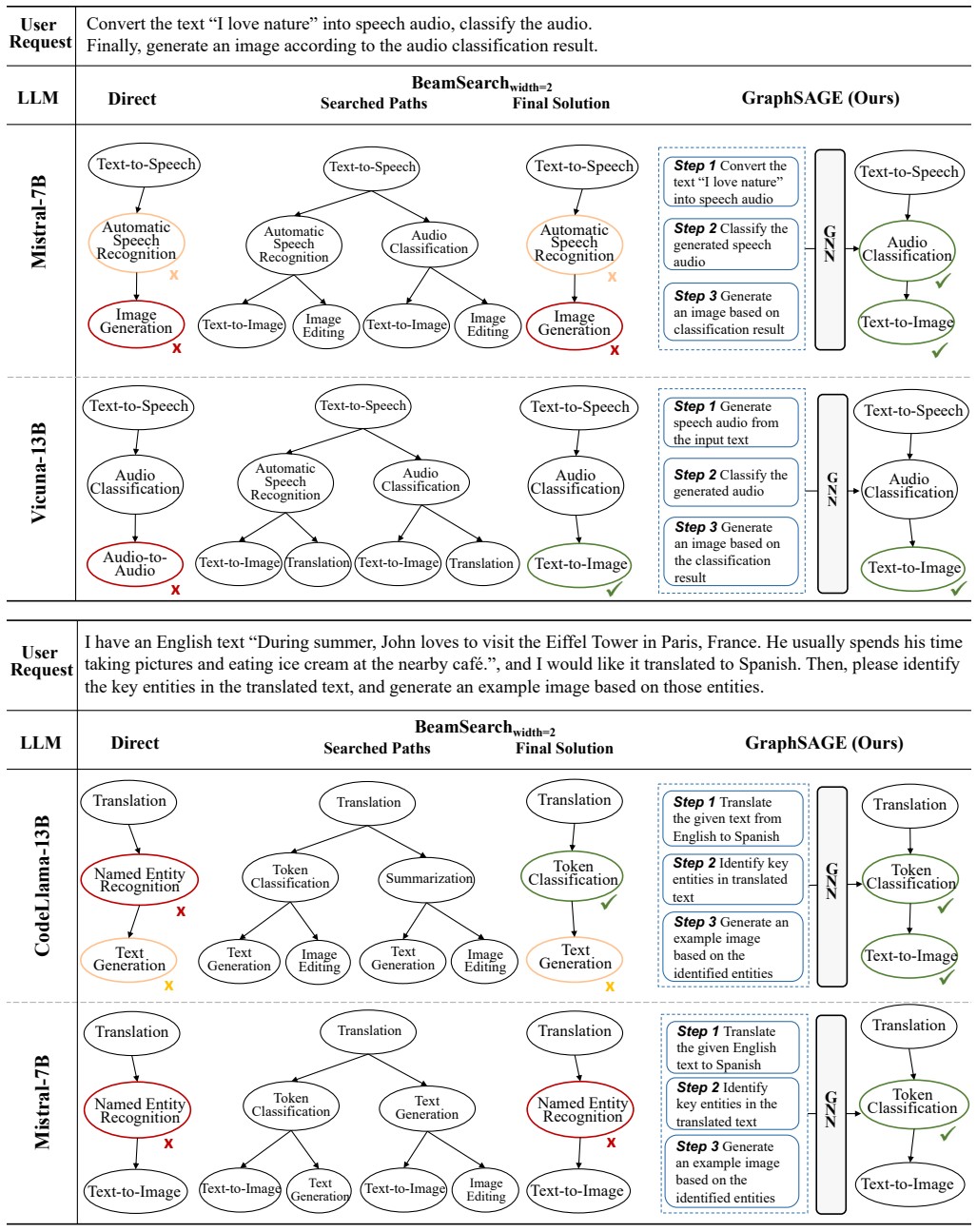

Figure 9: **Case Study of GNN's Effectiveness**. Nodes colored in pink and red denote wrongly predicted task or hallucinated task, respectively. Due to space limitations, we only show the first four valid searched paths of BeamSearch for illustration. Even though LLM can explore ground-truth paths during searching, they **lack certain instruction-following and reasoning abilities to consistently choose the optimal path**. On the contrary, GNN can correctly align decomposed steps with suitable tasks, **hitting the ground-truth result**.

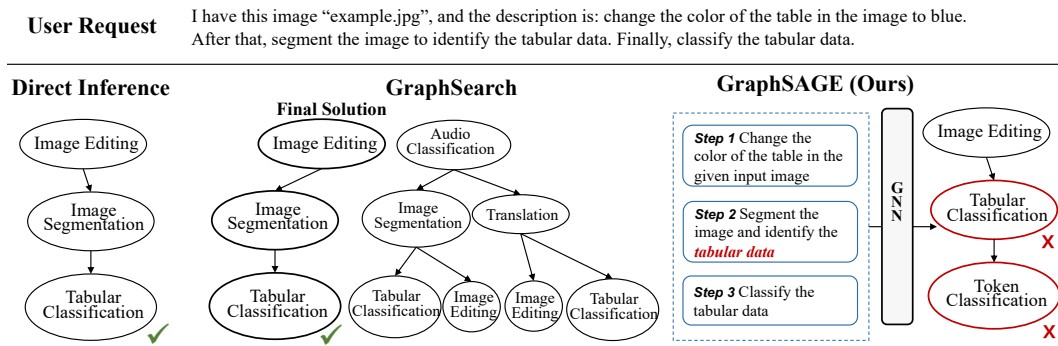

Figure 10: **Failure cases of GNN**. Our framework relies heavily on the quality of decomposed task steps. Ambiguous steps (step 2 which actually incorporates two steps) may mislead GNN to select the wrong task.

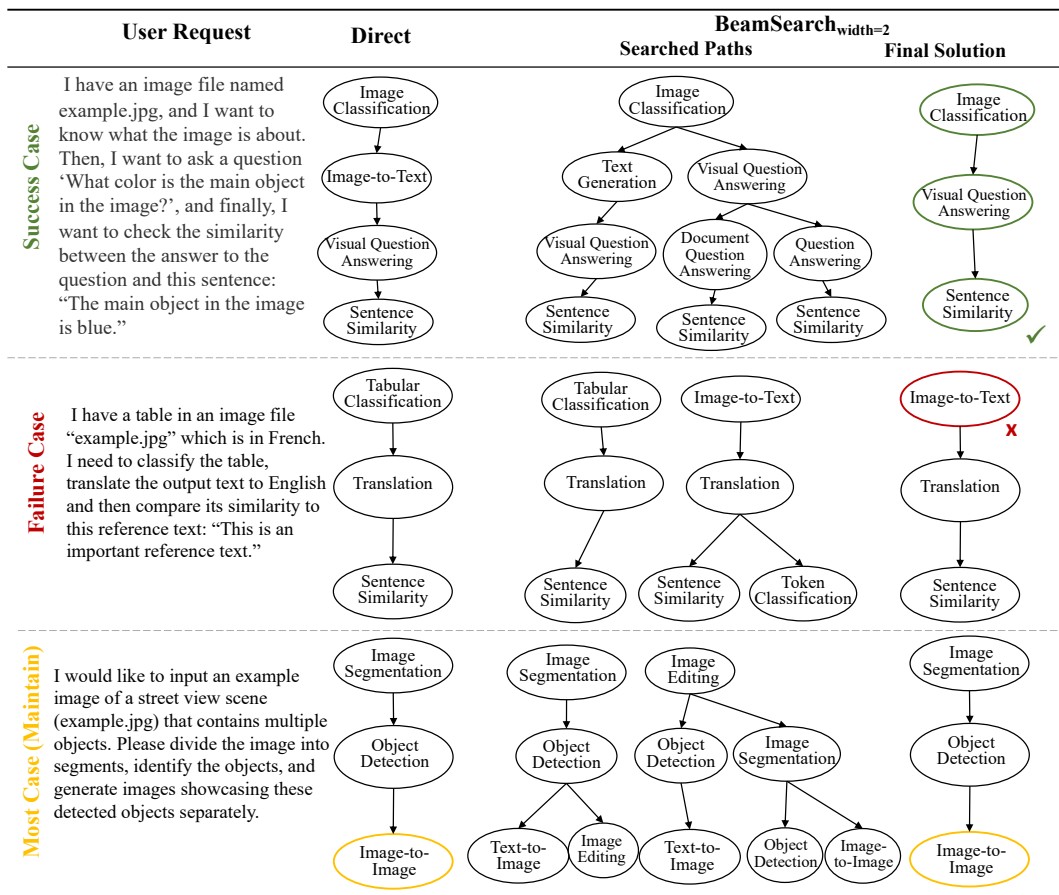

Figure 11: **Diagnosing GraphSearch for GPT-4**. We provide three types of cases: **Success Cases** where GPT-4 can leverage inherent reasoning abilities to select the optimal invocation path, which may even fix wrongly decomposed task steps. **Failure Cases** where GPT-4 miss in the extremely long context, containing the whole task graph, all searched paths, and instructions, selecting an unsatisfactory path. **Maintain Cases** where the searched result is the same as direct inference result, and those wrongly predicted tasks can still not be refined even under exhaustive search.

# I Broader Impacts

In this paper, we did not use any non-public data, unauthorized software, or API in our paper, there are no privacy or other related ethical concerns. Similar to other models designed for autonomous agents, our model also has the unfortunate potential to be used for malicious attacks. We pledge to restrict the usage of our model exclusively to the realm of research to prevent such misuse.

