# OpenReview forum: "Can Graph Learning Improve Planning in LLM-based Agents?"
_NeurIPS.cc/2024/Conference — NeurIPS 2024 poster_

### Official Review · Reviewer_wd1Z · 2024-06-23

**Soundness:** 2
**Presentation:** 2
**Contribution:** 2
**Rating:** 5
**Confidence:** 4

**Summary:**

This work proposes to investigate the integration of LLMs and GNNs for task planning. Specifically, LLMs are employed for the request decomposition stage while GNNs are employed for the task retrieval stage. Experiments demonstrate gains on various datasets across various LLMs.

**Strengths:**

+ the intersection of LLMs and graphs is interesting
+ task planning is an important problem

**Weaknesses:**

- It would be nice to present a figure/clear description of how LLMs and GNNs are integrated for task planning, somewhere in Section 4. For now it seems that Sec 4.2&4.3 focus on details of existing GNN proposals, which is not the technical contribution of this work, so it might make more sense to better highlight the unique way of composing them.

- There is some literature on combining LMs and GNNs for various purposes, most related ones include [1-3]. I'm not sure if the related work discussion is currently adequate and perhaps it would be helpful to start from [1-3]. In addition, maybe some of these could be valid baselines, especially for works like [2] and [3].

- LLMs are used in stage 1: decomposition while GNNs are used in stage 2: task retrieval, right? One might say that the decomposition step is a more "planning" process than just retrieving task descriptions with GNNs, so it might make sense if the proposal could incorporate GNNs in stage 1. This decoupled/independent application of LLMs and GNNs each in one separate step seems less exciting, compared to the deeper combinations of LLMs and GNNs like [1-3]. Perhaps if the authors could empirically show that keeping them separate is better than these existing proposals?

- It would be nice to show qualitative analysis for the improved stage 2, highlighting at least one killer example in the main paper where LLMs failed task retrieval but GNNs helped. Totally understand the space limit so merely a suggestion.

- There is a mismatch between Sec 3.1 and 3.2. While 3.1 is empirically looking at LLM graph reasoning specific to the planning task, Sec 3.2 is looking at more general interpretations of LLM graph reasoning in natural language. As the title suggests, the main focus of this work is specifically task planning, so I'm not sure if 3.2 is integral/significant to the main argument and could be a standalone work.

- Minor point, not sure if Figure 2 is presenting the strongest models both for open and proprietary LLMs. I see that GPT-4 is employed in Section 5.1, why not here?

- Minor point, but the title could be a bit vague. "Graph learning" is a broad term, could mean the GNN-centric line of graph representation learning and could also be referring to the more recent LLM&graphs research direction. Maybe better highlight that this work presents a way to combine LLMs and GNNs, that they execute different stage of the task planning pipeline.

[1] Yasunaga, Michihiro, et al. "Deep bidirectional language-knowledge graph pretraining." Advances in Neural Information Processing Systems 35 (2022): 37309-37323.

[2] He, Xiaoxin, et al. "Harnessing explanations: Llm-to-lm interpreter for enhanced text-attributed graph representation learning." The Twelfth International Conference on Learning Representations. 2023.

[3] Perozzi, Bryan, et al. "Let Your Graph Do the Talking: Encoding Structured Data for LLMs." arXiv preprint arXiv:2402.05862 (2024).

**Questions:**

please see above

**Limitations:**

yes

---

> ### Author Rebuttal · Authors · 2024-08-06
>
> We sincerely thank the reviewer for your insightful and constructive feedback.
>
> ---
>
> > Weakness 1: Methodology figure
>
> Thank you for your suggestion. The methodology figure is provided in Figure 6 of original paper and better illustrated in **Figure 2 of the PDF in the global response**. We will relocate it to the main text for better clarity.
>
> > Weakness 2: Comparison to related works [Ref A-C]
>
> Thank you for pointing out these works. We will provide a detailed discussion in the revised manuscript. Our approach differs in both application and methodology: DRAGON [A] pre-trains a language-knowledge model for QA tasks, TAPE [B] focuses on node classification on text-attributed graphs, and GraphToken [C] addresses basic graph reasoning tasks. Unlike these papers, task planning involves a complex and ambiguous user request, which encompasses multiple tasks and thus does not belong to any applications in [A-C]. Here is a detailed comparison:
> | Paper          | Application        | Graph Type                 | Methodology             |
> | -------------- | ------------------- | -------------------------- | -------------------------------------- |
> | DRAGON [A]     | Question Answering  | Knowledge Graph            | LM                                     |
> | TAPE [B]       | Node Classification | Text-attributed Graph (TAG)      | LLM → LM → GNN |
> | GraphToken [C] | Graph Reasoning     | Graph w/o. Text            | GNN → LLM                  |
> | Ours           | Task Planning for Language Agents      | Task Graph (a kind of TAG) | LLM → LM + GNN             |
>
> Then, we considered using these as baselines. First, when language agents are deployed to a new environment, there is no training data available and training-free algorithms are needed (Table 1), making the above approaches inapplicable. Second, in the training-based scenario (Table 2), [B] and [C] can be adapted to our setting as baselines. Specifically, we adapted TAPE for task planning by mapping user requests to task labels using its original LLM-LM-GNN architecture. For GraphToken, we treated the user request as the input question, the expected plan as the answer, and the GNN encoded the task graph into latent embeddings.
>
> **Results in Table 2 in the PDF file of the global rebuttal show our method consistently outperforms them**. TAPE is unsuitable for task planning as its classification approach simplifies task planning, overlooking task dependencies. While GraphToken indeed demonstrates superior performance over direct LLM inference, we have noted instances of minor hallucination. This observation suggests that GraphToken's understanding of the task graph is not yet perfect. In addition, GraphToken is limited to open-source LLMs.
>
> > Weakness 3: Rationale for the two-stage algorithm
>
> Thank you for providing valuable insights. The rationale is justified by three main reasons: 1) Training-free deployment is needed in new environments for language agents, whereas methods in [Ref A-C] require extensive training. 2) Flexibility to use proprietary LLMs, like GPT-3.5, while GraphToken [C] is limited to open-source LLMs. 3) Simplifying the methodology while integrating the strengths of both LLMs and GNNs. **To illustrate this advantage, we compare our method with [B, C] in the PDF and demonstrate superior performance.**
>
> Within the framework, both the LLM and GNN contribute to the planning. The LLM interprets ambiguous user requests into concrete steps, while the GNN selects suitable tasks aligned with those steps. In the second stage—task retrieval, which involves graph planning and reasoning—LLMs face challenges due to hallucinations and limited graph understanding abilities (see Sections 3.1 and 3.2). Our framework leverages the strengths of both LLMs and GNNs in task planning, ensuring more accurate planning results.
>
> > Weakness 4: Qualitative analysis for the improved stage 2
>
> We will move **Figure 9 on Page 24** to the main paper, where GNNs successfully correct paths that LLMs fail to generate.
>
> > Weakness 5: Mismatch between Section 3.1 and 3.2.
>
> We thank the reviewers for allowing us to clarify Section 3.2's role in task planning for language agents like HuggingGPT. Current research in this area mainly uses pre-trained LLMs and focuses on prompt design (e.g., [20,23,31,42]), but exploring their fundamental limits is essential. Section 3.2, while focused on general graph planning, is also relevant to task planning, as evidenced by our theorems. In existing studies, task graphs are often presented in Eq. 3, where the edge list is describe by natural language and the initial states are the task descriptions of each task node, detailed in Appendix A.9 of [30]. Theorem 1 shows a positive result that confirms Transformer models' expressiveness for such tasks. However, sparse attention, a inductive bias of language [48], limits expressiveness (Proposition 1) and auto-regressive loss could introduce spurious correlations (Theorem 2). These negative results inspire us to jointly leverage GNNs and LLMs for task planning to combine their strengths.
>
> > Weakness 6: Missing GPT-4 in Figure 2
>
> We omitted GPT-4 and CodeLlama-7B in Figure 2 due to limited space. The Task-F1 score of GPT-4-turbo is 77.60% with hallucination ratios of 0.12% for nodes and 6.73% for edges.
>
> > Weakness 7: Vagueness of "graph learning"
>
> Thank you for bringing our attention to the vagueness of the title. We categorize our work into the more recent LLM&Graphs research direction as we integrate GNNs to boost task planning for language agents. We will change the title in the revised manuscript.
>
> ---
>
> [Ref A] Yasunaga, Michihiro, et al. "Deep bidirectional language-knowledge graph pretraining." NeurIPS 2022.
>
> [Ref B] He, Xiaoxin, et al. "Harnessing explanations: Llm-to-lm interpreter for enhanced text-attributed graph representation learning." ICLR 2024.
>
> [Ref C] Perozzi, Bryan, et al. "Let Your Graph Do the Talking: Encoding Structured Data for LLMs." arXiv:2402.05862.

---

> > ### Comment · Reviewer_wd1Z · 2024-08-08
> >
> > I would like to thank the authors for their detailed response.
> >
> > - In the global comment, the authors claim that *this is the first attempt to theoretically understand the fundamental limits of LLMs in this application (or more broadly, in LLMs for graph problems)*. I wonder if [1] might weaken this claim.
> >
> > - Again "improving LLMs for planning" and "studying the fundamental limits of transformers/LLMs in graphs" are two separate research question. I'm not sure the value of having part #2 in this part #1-centric work and how it benefits planning.
> >
> > Other than that, reviewer xcxm do raise many valid points, especially how this work might be over-claiming on several fronts regarding existing literature and I share these concerns. I'm upping my score to 5 at the moment, but not sure if I'm comfortable championing without reviewer xcxm's response.
> >
> > [1] Dziri, Nouha, et al. "Faith and fate: Limits of transformers on compositionality." NeurIPS 2023

---

> ### Author Response · Authors · 2024-08-08
>
> Dear Reviewer,
>
> We greatly appreciate your insightful suggestions and comments, which have significantly contributed to the refinement of our paper. We are also thankful for your acknowledgment of our rebuttal efforts.
>
> Regarding the first question, we thank you for bringing this important reference to our attention and will discuss it in the revised manuscript. The issues investigated in [1]—Multiplication, Einstein’s Puzzle, and Maximum Weighted Independent Set on a sequence of integers—are not traditionally defined as graph problems because they do not involve graphs as inputs. Instead, they conceptualize the computations as computation graphs, similar to the computation graph taken by Pytorch.
>
> In response to the second question, our application takes a task graph and a user request in natural language as inputs, and produces a path on the task graph as the output. The ability to accurately interpret the task graph input is therefore crucial. Our experiments reveal that LLMs often hallucinate about non-existent nodes and edges, or the generation of disconnected paths. This suggests that (1) LLMs struggle to understand task graph inputs and (2) enhancing LLMs' graph comprehension could improve their application in task planning. Consequently, we explore the reasons behind LLMs' difficulties with task graph comprehension in Section 3.2.
>
> Regarding the over-claiming concerns raised by Reviewer xcxm, there is a field known as task planning in both traditional AI and language agents. While they share the same name, they are distinct areas. We did not realize that there is an area called task planning in traditional AI until it was highlighted by the reviewer. The references cited by Reviewer xcxm pertain to task planning in traditional AI, whereas our paper addresses task planning in language agents. Following the rebuttal to Reviewer xcxm, we will revise our title and statements to avoid confusion.
>
> If you have a different perspective, please feel free to share it with us, and we would be happy to address it during the discussion period.
>
> Warm regards,
>
> The Authors

---

### Official Review · Reviewer_xcxm · 2024-06-25

**Soundness:** 2
**Presentation:** 2
**Contribution:** 2
**Rating:** 5
**Confidence:** 3

**Summary:**

The paper proposes to use GNNs alongside LLMs for decision making problems represented in natural language and task graph inputs. Experimental results show that using GNNs provide improvements over just using LLMs.

**Strengths:**

The results highlight that the approach makes sense and work, with improvements across benchmarks.

The paper provides some theory albeit simple and straightforward: graphs encoded as input strings into transformers are limited in expressiveness.

**Weaknesses:**

The paper has some clarity problems, as both the problem statement and the methodology of the paper is not clear without several rereads.

1. The proposed definition of planning is vague and not well defined. The authors define task planning as a path finding problem on a task that fulfils user requests, where 'user requests' is not defined.

2. It is not clear anywhere how GNNs are used to improve the performance the task planning problem described in the paper. More specifically, the input to the models in Sec. 4 are unclear, i.e. what is the type of request and $v_i$ in (request, {s_1, \ldots, s_n}, {v_1, \ldots, v_n}) in line 231. Furthermore, it is not mentioned what the graph encoding of the inputs of the problem are. What are the nodes, edge and features of the input problem for use with the GNN? It appears later on that the datasets already give you the graphs. Make this clear if this is true and describe the graph encodings in one place.

3. "The task planning considered here is more flexible and cannot be directly translated into formal languages" is not explained. Both PDDL planning and the task planning problem the authors both have limitations as they require structured model inputs (PDDL language and graphs, respectively). Furthermore, the second half of the claim "cannot be directly translated into formal languages" is given no justification and explanation and appears as a handwavy argument to not consider these fields.

4. GNN and graph representation learning have been employed in task planning and decision making for several years so it is wrong to claim that the paper "presents an initial exploration into graph-learning-based approaches for task planning as pointed out in Section A.3. Related work concerning GNNs for PDDL planning is also missing, see e.g.
- Sam Toyer, Sylvie Thiébaux, Felipe W. Trevizan, Lexing Xie: ASNets: Deep Learning for Generalised Planning. J. Artif. Intell. Res. 68: 1-68 (2020)
- William Shen, Felipe W. Trevizan, Sylvie Thiébaux: Learning Domain-Independent Planning Heuristics with Hypergraph Networks. ICAPS 2020: 574-584
- Simon Ståhlberg, Blai Bonet, Hector Geffner: Learning General Optimal Policies with Graph Neural Networks: Expressive Power, Transparency, and Limits. ICAPS 2022
- Jiayuan Mao, Tomás Lozano-Pérez, Joshua B. Tenenbaum, Leslie Pack Kaelbling: What Planning Problems Can A Relational Neural Network Solve? NeurIPS 2023
- Dillon Ze Chen, Sylvie Thiébaux, Felipe W. Trevizan: Learning Domain-Independent Heuristics for Grounded and Lifted Planning. AAAI 2024

Other comments and suggestions:
- GNN definition not referenced, (1) is restrictive due to concatenation of edge features in aggregation function, but there are many GNNs that deal with edge features in other ways.
- (1) is an MPNN
- Define "\oplus is an operator" more rigorously. A mathematical operator on integers, floats, booleans? What is the XOR operating on, the bit representation of a number, or a cast of a number to a boolean?
- Figure 2: typos in legend. hallucination ratio not defined nor referenced anywhere in the text
- Theorem 1: assumptions should not be listed in the appendix, and instead should be given before the statement of the theorem
- F1-scores do not measure accuracy

**Questions:**

1. What is meant by "fulfils the users' requests"? Is this a classification problem, or a constraint satisfaction problem?
2. What is the explicit graph encoding (nodes, edges, features) of the problem for use with the GNNs?
3. Why are RestBench results only shown the no training results (Table 1) and not Table 2?

**Limitations:**

The authors addresses some limitations.
- The work requires graphs to be part of the input problem.
- Experiments repeats are not performed and error bars are not reported. Many score have marginal difference (e.g. <0.1 F1 score difference), yet this is not considered when highlighting the best performing model in the tables.

---

> ### Author Rebuttal · Authors · 2024-08-06
>
> We sincerely thank the reviewer for the insightful and constructive feedback.
>
> ---
>
> > Weakness 1, 3, 4 and Question 1: Definition of task planning and user request; Task planning and PDDL; Related work concerning GNNs for PDDL
>
> This paper investigates task planning for language agents, a new and increasingly important application area with the development of LLMs [30, 14, 43].
>
> **Definition of Task Planning for Language Agents:** Task planning involves a task graph, a directed graph where nodes represent tasks and edges indicate dependencies between tasks. Each node contains text detailing the task's name and usage in natural language. **The inputs are this task graph and a user request expressed in natural language**. The request is often ambiguous and personal, making it difficult to define mathematically. The output is a sequence of tasks and their order of invocation (a path on the graph) to fulfill the user's request.
>
> **Example:** Figure 1 in the PDF illustrates task planning for HuggingGPT [31]. Task nodes correspond to APIs from the HuggingFace website, such as "Translation", accompanied by text descriptions like "Translation is the task of converting text from one language to another." The **user request** is "Please generate an image where a girl is reading a book, and her pose matches the boy in 'example.jpg', then describe the new image with your voice." The output is a sequence of four APIs (nodes): {"Pose Detection", "Pose-to-Image", "Image-to-Text", "Text-to-Speech"}, outlining the execution order. By invoking these APIs on HuggingFace, user request can be fulfilled.
>
> Since the output is a sequence of tasks (a path on the task graph), task planning for language agents is similar to a constraint satisfaction problem (CSP). However, it cannot be solved as a CSP because **both the task graph features and user requests are expressed in natural language**.
>
> Unlike classic planning with fixed goals (e.g., arranging tiles in numerical order in the n-puzzle), task planning for language agents involves **diverse and open-ended goals** due to varied and personal user requirements. For example, on HuggingFace, user intentions span video, text, and image domains. Formulating such specific and context-rich requests into PDDL is quite difficult, as shown by the example request.
>
> To our knowledge, this is the first paper to explore the GNNs in task planning for language agents. We consider the references suggested by the reviewers to be important and constructive. Accordingly, we will discuss these references and narrow our title and statements to language agents in revised manuscript.
>
> > Weakness 2 and Question 2: Details of Framework
>
> Due to rebuttal space limitations, please refer to Weakness 5 of Reviewer ZJrg for the answer.
>
> > The theory is straightforward
>
> Thank you for helping us refine the theoretical contributions. Section 3.2 explains that LLMs task planning issues arise from language pretraining and auto-regressive loss biases, rather than the input format.
>
> In task planning for language agents, the existing studies focus on prompt engineering for pretrained LLMs, and we seek to investigate fundamental limits. We initially thought that the expressiveness was hindered by the input format. However, we discovered that the input format does not impede expressiveness, even when the hidden dimension is small (Theorem 1). Pursuing this line of inquiry, we found that sparse attention, an inductive bias inherent in language pretraining [48], restricts expressiveness (Proposition 1), and that the auto-regressive loss introduces spurious correlations (Theorem 2). These negative results inspire us to apply GNNs to complement LLMs.
>
> > Question 3: No RestBench results in Table 2
>
> As mentioned in Line 289-290, RestBench has only 100 data samples, which limits effective model training.
>
> > (1) is an MPNN and define "\oplus is an operator" more rigorously
>
> We use the GNN definition from [49, 50], which is a general formulation of message-passing-based GNN (e.g., GCNs or GATs). As Aggregate$^{(k)}$ can be any functions operating on the set, it can represent commonly used operations on edges, such as multiplication.
>
> We agree with the reviewer that it is a message-passing-based GNN and we will change its name in the revised manuscript. The input type of $\oplus$ depends on the type of Answer[k − 1][j] and c[i][j] and can be any operator.
>
> > Definition of hallucination ratio
>
> As mentioned in Line 122-124, it is defined as the percentage of non-existent nodes or links outputted by LLMs. Since LLMs' output domain is open-ended, their output may contain nodes or edges not present in the graph. Specifically, for a prediction {$v_1, \ldots,  v_n$}, the node hallucination ratio is computed as $\frac{ \sum_{i=1}^n \mathbb{I}(v_i \notin V)}{n}$ where $\mathbb{I}(\cdot) \in ${0, 1} is an indicator function. The same method applies to the link hallucination ratio.
>
> > Assumptions should be put in main text
>
> Due to space limitations, we initially included the assumptions in the appendix. We totally agree with the reviewer that these assumptions would be more appropriately placed in the main text. We will revise the manuscript accordingly.
>
> > Missing accuracy
>
> Thank you for your suggestion. **We have supplemented our results with accuracy metric in Table 1 of the PDF file in global rebuttal.** The results show that integrating GNNs significantly improves accuracy.
>
> > The work requires graphs to be part of the input problem
>
> Thank you for pointing out the limitation. In task planning for language agents, the dependencies of tasks naturally form a graph, which we believe is straightforward to obtain.
>
> > Error bar
>
> We followed previous works HuggingGPT [31] and TaskBench [30] to give the experimental results. The original papers did not provide performance variance as some experiments are extremely time-consuming and the variation is small. We will include error bars in the revised manuscript.

---

> > ### Comment · Reviewer_xcxm · 2024-08-08
> >
> > We thank the authors for their response and answers in response to the weaknesses and questions I pointed out in the review, namely
> >
> > (1) clarity of the problem statement and technical details in the methodology, and
> >
> > (2) ambiguity of the definition of "task planning", namely whether the problem is concerned with natural language or symbolic problem statements.
> >
> > The authors sufficiently address these weaknesses in their rebuttal, alongside the other minor comments, and I trust that authors are able to address them in the final version of the paper.

---

> ### Author Response · Authors · 2024-08-08
>
> Dear Reviewer,
>
> Thank you for your thorough review and insightful suggestions. Your comments have been invaluable in refining our paper.
> We appreciate your positive reception of our rebuttal and are glad that our response has addressed your concerns. We are committed to carefully revising the manuscript according to your feedback.
>
> Thank you again for your constructive review.
>
> Warm regards,
>
> The Authors

---

### Official Review · Reviewer_ZJrg · 2024-07-11

**Soundness:** 3
**Presentation:** 2
**Contribution:** 3
**Rating:** 6
**Confidence:** 3

**Summary:**

The paper proposes a GNN+transformers method to address some theoretical and practical issues in graph planning such as hallucinations. The task consists of matching a prompt that expresses a number of sub-tasks to a larger graph of tasks (the task pool) an agent (e.g., an LLM) can solve and invoke the right procedure to solve it (in the spirit of HuggingGPT).

**Strengths:**

The general problem is well formulated and the motivation behind GNNs to improve transformers on graph planning easy to follow. Section 3.1 (but not Figure 2) do a good job at describing the hallucination issue and the potential mitigations introduced by GNNs (my understanding is because they have grounded access to the true task graph).
Results suggest that GNNs improve performance on task planning, both in training and training-free regime.

**Weaknesses:**

Figure 2 is problematic. It is never referenced in the paper and there is no reference to the dataset used and how the task is specified (I assume is the same setting as HuggingGPT, but just because it appears in the same section).

I don’t follow the argument right after Eq. 3. Such representation may not be the standard representation of a graph, but it’s sufficient to fully reconstruct vertices and relationships (and is, in some cases, optimal, as pointed out by some works that you cite in the second paragraph of the introduction).

While I understand Proposition 1 and its demonstration by contradiction in the Appendix, one can always choose a larger Transformer that could attend all the instances in the DP problem. Furthermore, you write that the information accessible by the Transformer is O(logn), with n left undefined (it is the number of tokens?), and compared to |V|, the number of nodes in the graph. It would help to use the same control variable for a comparison.

I am not convinced by Theorem 1 and the following example 1. Despite a model never sees a path to d from a in the training set, that is not a proof (nor can be) that an LLM cannot interpolate (what some researcher in literature refers with the spurious term “emergent abilities”) from abc to bcd from other similar examples. As you correctly mention after Example 1, humans can concatenate abc and bcd. The ability to “correlate” can be learnt at training time by seeing different examples.

It is not totally clear to me, after reading the paper twice, how the representation of the GNN is combined with the LLM. That should be stated clearly and maybe integrated with Figure 1, which is not very illustrative. I’d like to see a formula that precisely describes the input of the problem (I guess a graph), how it is manipulated by the GNN and then fed to the LLM. I found an informal description of this step at lines 220-222, but I recommend to move that up in the paper.

**Questions:**

See each point in weaknesses, especially the third, fourth and fifth paragraphs.

**Limitations:**

See previous points.

---

> ### Author Rebuttal · Authors · 2024-08-06
>
> We sincerely thank the reviewer for your insightful comments and recognition of this work, especially for the motivation to adopt GNNs in this new application.
>
> ---
>
> > Weakness 1: Figure 2 is never referenced
>
> We apologize for the confusion regarding Figure 2. The experimental settings are the same as HuggingGPT [31], using the HuggingFace dataset [30], and tasks are APIs on HuggingFace. We will provide a clearer caption and reference it in the revised manuscript.
>
> > Weakness 2: Representation of graph in Eq. 3
>
> Thank you for helping us clarify Eq. 3. The edge list input format is a standard practice in works involving LLMs for graphs (e.g., [10, 41]). In existing studies, task graphs are often presented in Eq. 3, where the edge list is described by natural language and the initial states are the task descriptions of each task node, detailed in Appendix A.9 of [30]. Therefore, we adopt it as the basic graph representation for LLMs.
>
> > Weakness 3: One may choose larger Transformers and definition of O(logn)
>
> We thank the reviewer for the meticulous review. In task planning, the use of pre-trained LLMs is common, but these models often have a fixed size, while the input graph size can vary. In studying the expressiveness of Transformers, it is typically assumed that the Transformer's embedding size remains constant relative to the problem size, as referenced in [Ref B]. Regarding the variable $n$, we define it as the number of tokens, with $n=\Theta(\text{poly}(|V|))$. Each parameter in the Transformer is designed to store information up to O(logn) bits to reflect practical machine precision (e.g., float16 or float32). We will include this explanation in the revised manuscript for better clarity. We welcome any different perspectives and are open to discussing them further.
>
> > Weakness 4: Theorem 2 and Example 1
>
> The difficulty in path concatenation arises from the nature of the next-token-prediction loss itself--learning concatenation results in a higher training loss: When predicting the next token with the current node $i$ and target node $j$, the distribution of the next token that minimizes training loss should follow the corresponding distribution in the training dataset, i.e., $\Pr[\text{output} = k \mid \text{current node} = i \text{ and target node} = j] = \frac{N_{j,i,k}}{N_{j,i}}$. Path concatenation will alter the distribution from ${\frac{N_{j,i,k}}{N_{j,i}} \text{ for all } k}$, incurring a higher training loss.
>
> **Example:** Assume that the task is path-finding and training dataset contains the following three paths "a b a b", "b d b c d", and "a e a d e", where the first token is the source, the second token is the target, and the remaining tokens are a path from source to target. Now, suppose the input sequence is "a e a", if the model outputs d with probability 1, the cross-entropy loss is 0 (as only "a e a d e" in training data). However, if the model has the ability for path concatenation, the next token b will have a non-zero probability (as we have the path "a b c d e" through concatenation), and the cross-entropy loss will be larger than 0.
>
> Interestingly, after the submission, we find that this toy example has some practical implications in composition reasoning: if LLMs know the reasoning chain from $a\rightarrow b \rightarrow c$ and the chain $b\rightarrow c \rightarrow d$, they cannot perform reasoning $a\rightarrow b \rightarrow c \rightarrow d$ through path concatenation. This holds for existing LLMs including GPT-4 [Ref A].
>
> > Weakness 5: how GNN is combined with LLMs.
>
> **Two-stage Framework (Figure 2 in the PDF of global response or Figure 6 of original paper)**  First, LLM interprets user request into several concrete steps as {$ s_1, \ldots, s_n$} where $s_i$ is the $i$-th decomposed step. For the example request "Please generate an image where a girl is reading a book, and her pose matches the boy in 'example.jpg', then describe the new image with your voice.", the decomposition by GPT-4 is {"1. Analyze the pose of the boy", ... "4. Convert the generated text into audio"}. Then, we leverage GNN for task retrieval, sequentially matching each step description $s_i$ to a suitable task $v_i \in V$ (e.g., matching "1. Analyze the pose of the boy" to "Pose Detection"), thus generating the invocation path.
>
> Then we provide the detailed GNN encoding process:
>
> * **Task Graph $G=(V,E,X)$** Each node $v \in V$ represents an available task with a text $t_v$ describing its function (e.g., "Translation. Translation is the task of converting text from one language to another.").  Each edge $(u,v) \in E$ indicates a dependency between tasks (e.g., the output format of task $u$ matches the input format of task $v$). We use an LM (i.e., e5-355M) to generate initial node features as $x_v = \text{LM}(t_v)$.
> * **GNN Encoding** GNN incorporates dependencies between tasks and refines task embeddings as $h_v = \text{GNN}(x_v, G)$ for node $v$. We then leverage updated embeddings for task retrieval: each step can be represented as $x_{i}^{\text{step}} = \text{LM}(s_i)$, we compute the dot product between $x_{i}^{\text{step}}$ and task embeddings {$h_v$} for task selection.
>
> Finally, we explain GNN training. Each data sample in datasets contains the user request, a sequence of decomposed steps  {$s_1, \ldots, s_n$} for the request, and ground-truth task invocation path {$v_1, \ldots, v_n$}. We collect each $(s_i, v_i)$ pair and use a Bayesian Personalized Ranking loss for GNN training to realize its retrieval ability.
>
> Thank you for pointing out the confusion, and we will improve the technical presentation accordingly.
>
> ---
> [Ref A] Boshi Wang, et al. "Grokked Transformers are Implicit Reasoners: A Mechanistic Journey to the Edge of Generalization." arXiv:2405.15071.
>
> [Ref B] William Merrill, and Ashish Sabharwal. "The parallelism tradeoff: Limitations of log-precision transformers." TACL 2023.

---

### Official Review · Reviewer_s7yo · 2024-07-13

**Soundness:** 3
**Presentation:** 3
**Contribution:** 3
**Rating:** 7
**Confidence:** 3

**Summary:**

The paper formulates task planning as a graph decision-making problem. Through theoretical analysis, the paper shows the biases of attention and auto-regressive loss impede LLMs’ ability to effectively navigate decision-making on graphs. To mitigate this, the paper proposes to integrate GNNs with LLMs. Experiments demonstrate it can enhance overall performance.

**Strengths:**

1. The paper provides a solid theoretical analysis of LLMs understanding graph-structured inputs, which is inspiring for future works.
2. The proposed method is effective for task planning.
3. The paper is well-written and easy to follow.

**Weaknesses:**

1. Combining LLM with GNN for graph-related tasks has been studied in many previous works and is not very novel.

**Questions:**

N/A

**Limitations:**

Yes

---

> ### Author Rebuttal · Authors · 2024-08-06
>
> We sincerely thank the reviewer for your insightful comments and recognition of this work, especially for acknowledging our theoretical contributions and superior performance.
>
> ---
>
>
> > Combining LLM with GNN for graph-related tasks has been studied in many previous works and is not very novel.
>
> Thank you for helping us to clarify this point.
>
> Unlike previous LLM+GNN works that often concentrate on text-attributed graphs or knowledge graphs, our research ventures into a new application, i.e., task planning for language agents. This area is gaining increasing attention with the development of LLMs [30,14,43]. Existing studies in this domain have centered around optimizing prompt designs for pre-trained LLMs. This paper initiates the exploration of graph learning-based methodologies within this area. Motivated by theoretical analysis, we propose a tailored approach to combine GNNs and LLMs for task planning, which demonstrates significant performance improvements over existing baselines. **We have also added experiments to show that the proposed combination outperforms existing LLM+GNN solutions [Ref A, B] in Table 2 of the PDF in the global rebuttal**.
>
> ---
>
> [Ref A] He, Xiaoxin, et al. "Harnessing explanations: Llm-to-lm interpreter for enhanced text-attributed graph representation learning." ICLR 2024.
>
> [Ref B] Perozzi, Bryan, et al. "Let Your Graph Do the Talking: Encoding Structured Data for LLMs." arXiv:2402.05862.

---

> > ### Comment · Reviewer_s7yo · 2024-08-11
> >
> > Thanks for the reply. I will keep my positive score.

---

> > > ### Author Response · Authors · 2024-08-11
> > >
> > > We would like to express our sincere gratitude for the time and effort you dedicated to reviewing our manuscript. Your insightful feedback has strengthened the experiments of this paper. We truly appreciate your contributions to the improvement of our work.

---

### Author Rebuttal · Authors · 2024-08-06

# Global Rebuttal

We sincerely thank all the reviewers for your constructive feedback and recognition of this work, especially for acknowledging **the theoretical contributions** (Reviewer s7yo), **the motivations to adopt GNNs** (Reviewer s7yo and ZJrg), and **the superior performance** (Reviewer s7yo, ZJrg, and xcxm).

This paper focuses on a new application, i.e., task planning for language agents. This area is becoming increasingly popular and important with the development of LLMs [30, 14, 43]. Given its practical significance, there are many existing empirical works in this area while their focus is prompt design for pre-trained LLMs (e.g., [20,23,31,34,42]). We would like to re-emphasize the novelty and technical contributions of this work:

- To our knowledge, this is the **first attempt to theoretically understand the fundamental limits of LLMs in this application** (or more broadly, in LLMs for graph problems). We provide proofs that Transformers are expressive enough for planning on task graphs (Theorem 1), yet the attention mechanism's inductive bias could limit this expressiveness (Proposition 1), and auto-regressive loss might introduce spurious correlations (Theorem 2).
- Inspired by both theoretical and empirical findings, we propose to explore the use of GNNs in this application, **an orthogonal direction to existing studies**. Specifically, we propose an LLM+GNN solution that is flexible, supports **both training-free (zero-shot) and training-based algorithms**, and is compatible with **both open-sourced and closed-sourced LLMs**.
- Extensive experiments demonstrate the proposed solution outperforms existing methods on TaskBench and RestBench in both **performance and efficiency**. Furthermore, the proposed solution complements to existing prompt engineering and fine-tuning techniques, with performance further enhanced by improved prompts or a fine-tuned model.
- As an early exploration of applying GNNs to language agents, our research emphasizes that LLMs alone may not be sufficient, and traditional methods like GNNs have a role to play. We hope this work will inspire future research in language agents.


In line with the reviewers' constructive feedback, we have made the following additions and clarifications in our rebuttal:
- We adapted TAPE [Ref A] and GraphToken [Ref B] for task planning and included them as new baselines, with results provided in Table 2 of the PDF. (Response to Weakness 2 of Reviewer wd1Z)
- We clarified the definition of task planning for language agents and how it differs from traditional planning tasks. (Response to Weakness 1, 3, 4 of Reviewer xcxm)
- We provided more details on our proposed framework. To enhance readability and facilitate understanding, illustrations of task planning for language agents and our framework are included in Figure 1 and Figure 2 of the PDF file. (Response to Weakness 5 of Reviewer ZJrg)
- We provided results of an extra metric, Accuracy, for both baselines and our method in Table 1 of the PDF, demonstrating significant improvements across different datasets and LLMs. (Response to Reviewer xcxm)

Please don't hesitate to let us know of any additional comments on the manuscript or the changes.

---

[Ref A] He, Xiaoxin, et al. "Harnessing explanations: Llm-to-lm interpreter for enhanced text-attributed graph representation learning." ICLR 2024.

[Ref B] Perozzi, Bryan, et al. "Let Your Graph Do the Talking: Encoding Structured Data for LLMs." arXiv:2402.05862.

---

### Decision · Program_Chairs · 2024-09-25

**Decision:**

Accept (poster)

**Comment:**

The paper explores the combination of LLMs and GNNs for task planning. It presents theoretical insights into LLM limitations on graph-structured data and demonstrates empirical gains across various datasets. While reviewers appreciate the novelty and potential of the approach, they raise concerns about the clarity of presentation, theoretical arguments, and comparisons to existing literature. The authors have addressed some of these concerns in their responses.
I lean towards recommending acceptance for this paper. The idea is interesting and the results are promising, but there's room for improvement in clarity and positioning within the existing literature. The authors should use the feedback to strengthen the paper for the final version.